# Generation of three-dimensional meat-like tissue from stable pig epiblast stem cells

Gaoxiang Zhu[1,5], Dengfeng Gao[1,5], Linzi Li[2,5], Yixuan Yao[1], Yingjie Wang[1], Minglei Zhi [1], Jinying Zhang[1], Xinze Chen[1], Qianqian Zhu[1], Jie Gao[1], Tianzhi Chen [1], Xiaowei Zhang[1], Tong Wang[1], Suying Cao[3], Aijin Ma[4] ✉, Xianchao Feng[2] ✉ & Jianyong Han [1] ✉

Cultured meat production has emerged as a breakthrough technology for the global food industry with the potential to reduce challenges associated with environmental sustainability, global public health, animal welfare, and competition for food between humans and animals. The muscle stem cell lines currently used for cultured meat cannot be passaged in vitro for extended periods of time. Here, we develop a directional differentiation system of porcine pre-gastrulation epiblast stem cells (pgEpiSCs) with stable cellular features and achieve serum-free myogenic differentiation of the pgEpiSCs. We show that the pgEpiSCs-derived skeletal muscle progenitor cells and skeletal muscle fibers have typical muscle cell characteristics and display skeletal muscle transcriptional features during myogenic differentiation. Importantly, we establish a three-dimensional differentiation system for shaping cultured tissue by screening plant-based edible scaffolds of non-animal origin, followed by the generation of pgEpiSCs-derived cultured meat. These advances provide a technical approach for the development of cultured meat.

In recent years, global food consumption has increased due not only to the expanding worldwide population but also to a shift in eating patterns[1,2], including increased demand for foods of animal origin (e.g., meat), putting additional strain on the livestock industry[3]. These increases have had negative impacts on livestock[4], food security[5], animal welfare[6], climate change[1,7–9], environmental sustainability[10–12], public health[13,14], and competition between human and animal rations[1,15]. Moreover, livestock breeding is unable to meet the rising demand for meat protein[4]. As a consequence, it is essential to develop new food technologies and alternate ways to address the aforementioned concerns[15]. Cultured meat (CM) not only promotes environmental aims[16] but also has the potential to produce "clean meat"[17,18].

CM, a disruptive future food technology that efficiently blends clean production and sustainable food creation[19,20], uses stem cell technology to produce edible meat tissue through cell differentiation in vitro[20,21]. The pace of innovation in CM development has accelerated with the continuous development of cross-disciplinary approaches, but it still faces major technical challenges[19,22–24], including a lack of initiating cell lines and sufficient technologies for serum-free culture, scaffolding materials, large-scale suspension, and expansion culture. One of the cornerstones of the production of CM is a high-quality initiating cell line. Currently, accessible cell lines include mesenchymal stem cells and muscle stem cells (MuSCs), which can be amplified only a limited number of times before losing their capacity to differentiate[25]. While immortalized cell lines derived from bovine satellite cells[26] and chicken fibroblasts[27] have been reported as possibilities for addressing this shortcoming, the regulating mechanism remains to be elucidated. Unlike these cells, stable pluripotent stem cells (PSCs) derived from epiblasts may self-renew indefinitely and differentiate into any type of somatic cell, providing a new avenue for

[1]State Key Laboratory of Animal Biotech Breeding, College of Biological Sciences, China Agricultural University, Beijing, China. [2]College of Food Science and Engineering, Northwest A&F University, Yangling, Shaanxi, China. [3]Animal Science and Technology College, Beijing University of Agriculture, Beijing, China. [4]School of Food and Health, Beijing Technology and Business University, Beijing, China. [5]These authors contributed equally: Gaoxiang Zhu, Dengfeng Gao, Linzi Li. ✉e-mail: maaj@btbu.edu.cn; fengxianchao1@hotmail.com; hanjy@cau.edu.cn

the development of CM[28]. The establishment of stable PSCs from livestock species has been a challenge in the field for the last 40 years[19]. However, the generation of stable porcine pre-gastrulation epiblast stem cells (pgEpiSCs) provide a new opportunity[29]. In addition, it is essential to establish serum-free differentiation[30] of livestock PSCs to achieve three-dimensional differentiation in edible scaffolds and provide a theoretical basis and system reference for later cell large-scale expansion and tissue shaping culture.

In this study, we further validated the stability of the pgEpiSCs obtained in our previous work[29]. We also successfully established a myogenic differentiation system with a complete, transgene-free, and serum-free process and verified the characteristics of differentiated cells at the transcriptome and metabolome levels, ultimately achieving the development of CM derived from pgEpiSCs on an edible 3D scaffold without any animal components.

## Results

### pgEpiSCs are ideal initiating cells for the development of CM

We previously established a cell line (pgEpiSCs)[29] that can not only be sustained in long-term culture to achieve consistent clonal proliferation but also retain genome stability, both of which are considered ideal features of initiating cells for the development of CM.

To evaluate the pluripotency maintenance and differentiation potential of high-passage pgEpiSCs, we performed long-term cell culture in vitro and embryoid body (EB) differentiation assays and found that pgEpiSCs showed distinct and smooth colony boundaries and alkaline phosphatase (AP) activity (Supplementary Fig. 1a, b) and expressed major pluripotency protein markers (Fig. 1a) after 200 passages. We further confirmed the genomic stability of the pgEpiSCs at high passages by using an alkaline comet and karyotype assay because the capacity of PSCs to preserve their genetic integrity is critical to their ability to self-renew indefinitely. The results indicated that normal karyotypes and no significant accumulation of DNA damage occurred in the pgEpiSCs at low or high passages during the long-term passaging process (Supplementary Fig. 1d, g, i), while damaged DNA formed a comet tail in the $H_2O_2$-treated cells (Fig. 1b and Supplementary Fig. 1c). We also isolated porcine MuSCs (pMuSCs) from porcine muscle and investigated their characteristics during culturing in vitro. These results (Supplementary Fig. 2) showed a lower differentiation ability of pMuSCs at P13, indicating that pMuSCs lost the features and differentiation potential of MuSCs during long-term culture in vitro, consistent with a previous report[31]. However, pgEpiSCs at different passages could form EBs and differentiate randomly (Supplementary Fig. 1e, f), accompanied by down-regulation of core pluripotency-related genes and up-regulation of genes associated with differentiation into all three major lineages (Supplementary Fig. 1h), producing ectoderm, mesoderm, and endoderm tissue cells (Fig. 1c). These results suggest that the differentiation potential of pgEpiSCs is unaffected by increasing passage numbers.

Next, we performed RNA-seq for pgEpiSCs at passage 30 and at P200 to assess the properties of the cells more comprehensively. Unsupervised hierarchical clustering and correlation analysis revealed that pgEpiSCs at different passages could be clustered together (Supplementary Fig. 3a) and separated from porcine embryonic fibroblasts (PEFs) by principal component analysis (PCA) (Fig. 1d and Supplementary Fig. 3b). Additionally, the gene expression in pgEpiSCs was entirely different from that in PEFs (Fig. 1e), and pluripotency-related genes such as *OTX2*, *SOX2*, *NANOG*, *LIN28A* and *POU5F1* (*OCT4*) were highly expressed in pgEpiSCs at both low and high-passage numbers (Supplementary Fig. 3c–e). Gene Ontology (GO) biological processes enrichment analysis revealed the enrichment of terms associated with "ion transport" (Supplementary Fig. 3g, h), which are relevant to the function of PSCs. More significantly, our results indicated that neither cancer-related hallmark genes nor splicing factors were expressed in pgEpiSCs (Fig. 1f and Supplementary Fig. 3f), implying that no cancer

cell-related characteristics[32,33] were present in high-passage pgEpiSCs and that the ability to maintain long-term stable passage is associated with pluripotent gene expression. Overall, these results suggested that pgEpiSCs are ideal initiating cells for the development of CM due to their excellent long-term stable proliferation in vitro, differentiation potential, and maintenance of a stable genome.

### A serum-free approach for myogenic differentiation from pgEpiSCs

Since pgEpiSCs have limitless self-renewal capacity and multi-differentiation potential, we hypothesized that pgEpiSCs could be differentiated into muscle cells (MCs) by activating or inhibiting specific signaling pathways to establish a serum-free and transgene-free myogenic differentiation method. To initiate differentiation, we first applied the human or mouse differentiation system, in which the serum is replaced with insulin-transferrin-selenium. However, we found that the cells differentiated in this way were unable to adhere under feeder-free conditions (Supplementary Fig. 4a). Next, we adjusted the early differentiation system with modifications to the basal pgEpiSCs culture medium and found that B27 is important during initiating differentiation. We observed that the differentiating cells in the B27 group not only adhered better but also exhibited greater random differentiation potential (Supplementary Fig. 4b); therefore, we selected this medium for continued differentiation (Fig. 2a).

To further differentiate the cells towards paraxial mesoderm (MES), we designed small molecule combinations based on previous reports on MES-related signaling pathways[34–37], such as the WNT, BMP and TGF-β pathways. The results indicated that activation of WNT and inhibition of TGF-β is the best combination for initial induction of MES cells[34], in which MES-related markers (such as *T*, *PDGFRα*, and *MSGN1*) were highly expressed (Supplementary Fig. 4d–f, h), although the use of different molecule combinations did not affect the cell morphology (Supplementary Fig. 4c). We further confirmed pluripotency exit by AP staining and MES induction by immunostaining for T (Supplementary Fig. 4i, j). Interestingly, only the combination of WNT activation and TGF-β inhibition promoted further differentiation of myogenic progenitor cells (MPCs) (Supplementary Fig. 4g). The pgEpiSCs-derived MPCs expressed PAX7 in conjunction with MYOD (Fig. 2b, d), with a high proportion of CD31⁻CD45⁻CD56⁺ cells of 99.8% (Fig. 2c, e) and exhibited a differentiation efficiency similar with hPSCs[34] or mPSCs[38]. Moreover, we utilized N2 to perform serum-free terminal differentiation. The morphology of the differentiated cells was similar to that of cells induced with horse serum (HS)[35] (Fig. 2g). The pgEpiSCs-derived MCs showed that multiple spindle-shaped nuclei were arranged, suggesting multinucleated myotube formation, which was further confirmed by expressing markers of mature skeletal muscle fibers (SMFs), such as Myosin, MF20, and MYH3 (Fig. 2f–j and Supplementary Fig. 5a). The myofiber maturation was also indicated by high regulation of the expression of marker genes, such as *MYOG*, *MYMK*, *MYH2*, and *MYH3* (Fig. 2k) and the trends in gene expression associated with myogenic differentiation over time courses (Supplementary Fig. 6). In addition, the pgEpiSCs-MCs maintained high levels of cell viability and normal chromosomal numbers during differentiation (Supplementary Fig. 5b–d). Taken together, these results showed that we were able to consistently induce pgEpiSCs into mature SMFs by using transgene-free and serum-free differentiation protocols.

### Transcriptome features of myogenic differentiation of pgEpiSCs

We subsequently analyzed RNA-Seq data, with a focus on the expression levels of marker genes known to be involved in critical phases of myogenic lineage differentiation, to determine whether the differentiation scheme generates cells of the expected lineage at each stage. We identified 14546 genes with a median expression value TPM greater than 0.5 within each group (Supplementary Fig. 7a), and PCA demonstrated correlations among the three cell groups (Fig. 3a and

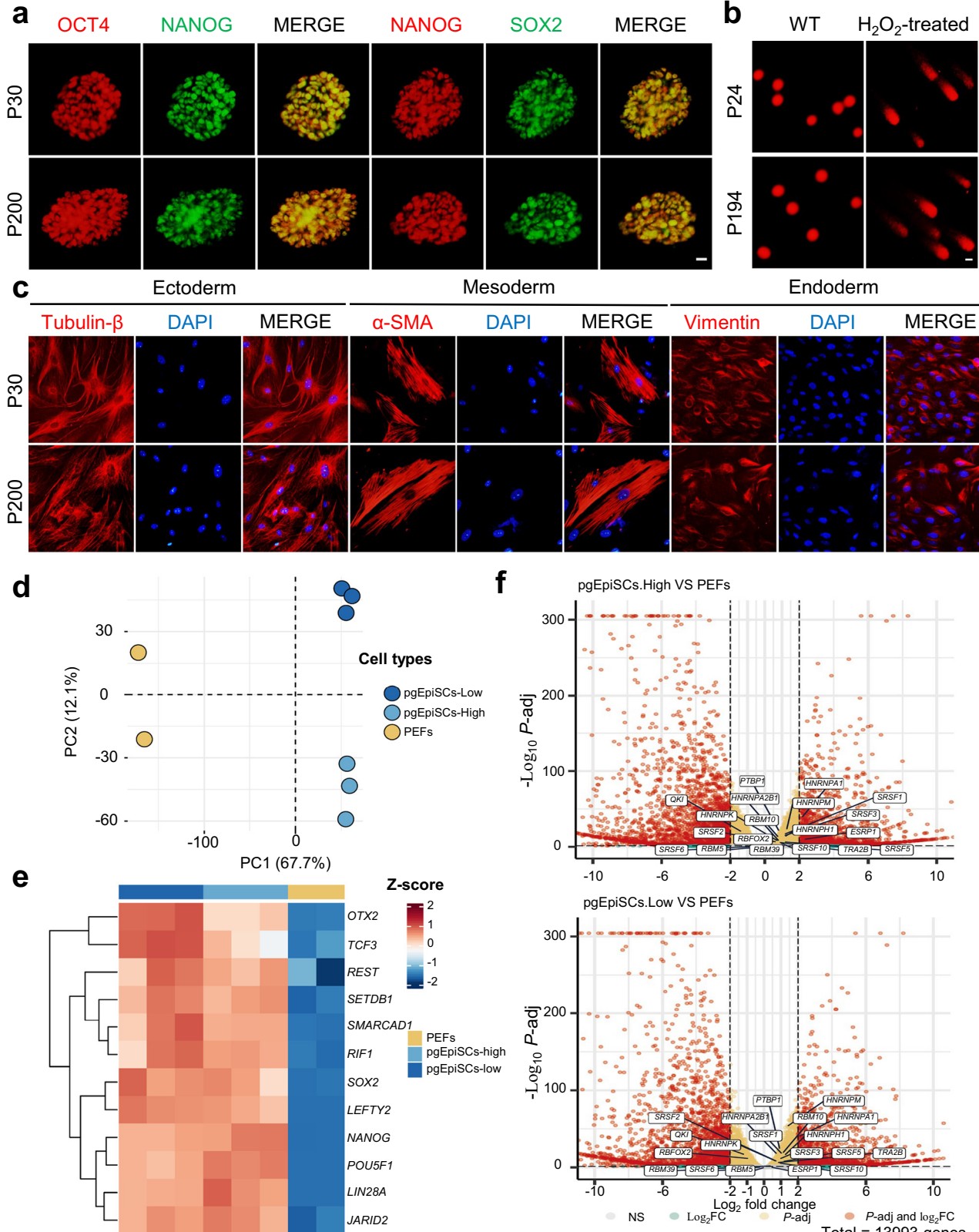

**Fig. 1 | Characteristics of pgEpiSCs at low and high passages.**
**a** Immunofluorescence staining of the pluripotency markers (OCT4, SOX2, and NANOG) in pgEpiSCs at low and high passages. Scale bar, 20 μm. **b** DNA damage levels in different generations of pgEpiSCs were evaluated by comet assay. $H_2O_2$ treatment was used as a positive control. Scale bar, 20 μm. **c** Random differentiation potential assay of pgEpiSCs at low and high passages in vitro. Immunofluorescence staining of Tubulin-β (ectoderm), α-SMA (mesoderm), and Vimentin (endoderm) in EBs derived from pgEpiSCs. DAPI was used to stain the nuclei. Scale

bar, 20 μm. **d** Principal component analysis (PCA) plot of RNA-Seq data from pgEpiSCs at low and high passages and PEFs. **e** Heatmap of differentially expressed genes (DEGs) for pluripotency of pgEpiSCs at low and high passages and PEFs. **f** Volcano plot of cancer-related splicing factors comparing pgEpiSCs and PEFs. Data were analyzed using the DESeq2 tool with the Wald-test. For (**a**) and (**c**–**f**), Low: P30; High: P200. For (**b**), Low: P24; High: P194. For (**a**–**c**), similar results were obtained in three independent experiments.

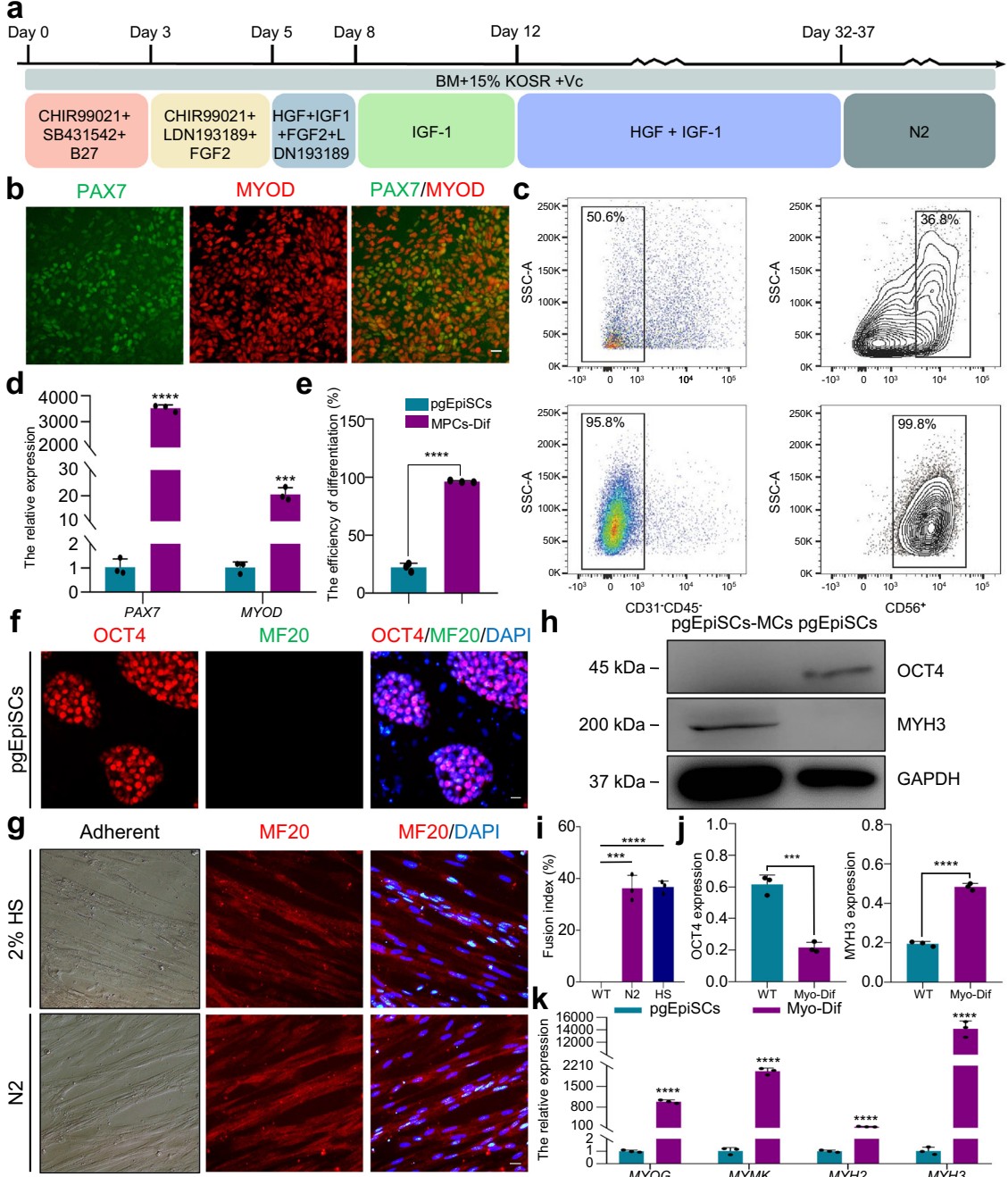

**Fig. 2 | Establishment of a serum-free culture system for myogenic differentiation of pgEpiSCs. a** Schematic of the myogenic differentiation of pgEpiSCs using a serum-free culture system. KOSR KnockOut™ Serum Replacement, B27 or N2 a serum-free additive with defined composition, CHIR99021 an activator of WNT/β-catenin or GSK-3α/β inhibitor, SB431542 an inhibitor of TGF-β, LDN193189 an inhibitor of BMP, FGF2 basic fibroblast growth factor 2, IGF insulin-like growth factor 1, HGF hepatocyte growth factor. **b** Immunostaining of PAX7 and MYOD in MDM III. **c** Flow cytometric analysis of the differentiation efficiency of pgEpiSCs-MPCs in MDM III. The CD31-CD45-CD56+ population was defined as pgEpiSCs-MPCs. **d** Analysis of *PAX7* and *MYOD* expression in MDM III. **e** The histogram characterizes the differentiation efficiency and correlates with (**c**). **f** Immunostaining of OCT4 (red) and MF20 (green) in pgEpiSCs as a negative control. Similar results were obtained in three independent experiments. **g** Cell morphology and immunostaining of mature muscle fibers from pgEpiSCs after treatment with 2% HS or N2. DAPI was used for nuclear staining. **h** Western blot analysis of OCT4, MYH3, and GAPDH expression in myogenic differentiation of pgEpiSCs. **i** The fusion index of mature muscle fibers from pgEpiSCs after treatment with N2 or 2% HS. **j** Quantification of OCT4 and MYH3 expression normalized to GAPDH from western blot gel. **k** Expression of genes related to the maturation of myogenic differentiation and skeletal muscle fibers. For (**b**, **f**, **g**), scale bar, 20 μm. For (**d**, **e**, **i**, **j**, **k**), error bars indicate means ± SD, *n* = 3. *** *p* < 0.001, **** *p* < 0.0001, similar results were obtained in three independent experiments and represent significant using two-tailed student's *t* test. Exact *P* values are listed in Source Data Fig. 2. WT undifferentiated pgEpiSCs, MPCs-Dif pgEpiSCs differentiated into MPCs, Myo-Dif terminally differentiated cells after N2 treatment.

Supplementary Fig. 7b), showing the changes in transcription from pgEpiSCs to pgEpiSCs-MPCs and pgEpiSCs-MCs. This finding was also corroborated by ternary plots (Fig. 3b) and the distribution of marker genes ranked by loading score (Fig. 3c).

To understand the myogenic differentiation process of pgEpiSCs in vitro, we utilized K-means clustering to divide genes with significant time-dependent changes in expression into ten clusters (Supplementary Fig. 7c, d). The signature genes of each cluster were vastly

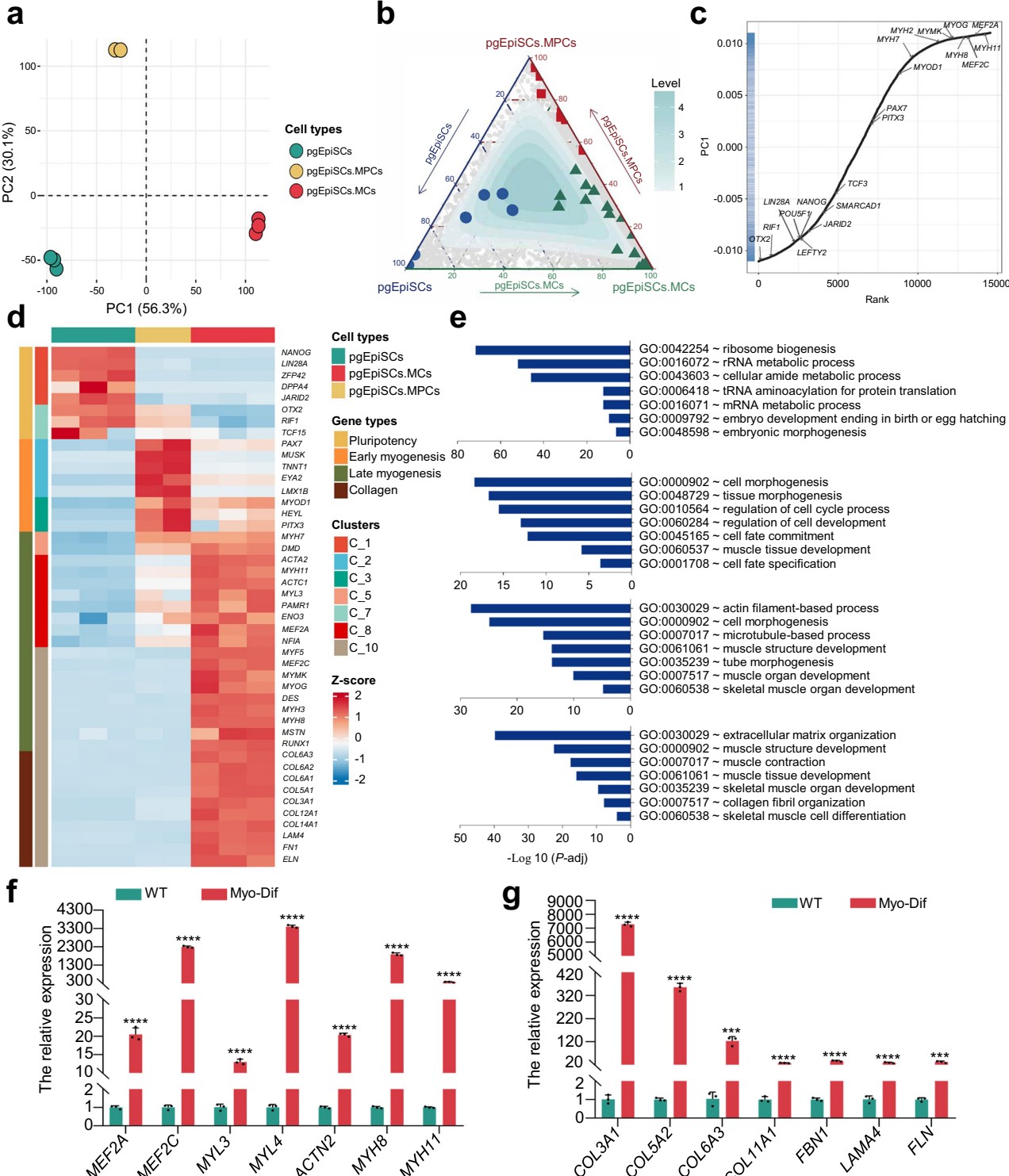

**Fig. 3 | Clustering analysis of differential gene expression patterns during myogenic differentiation of pgEpiSCs. a** PCA plot of three populations during the myogenesis of pgEpiSCs in vitro. Colors indicate different cell populations. **b** Ternary plot of three populations during the myogenesis of pgEpiSCs in vitro. Key markers of different cell populations are indicated with different colors. **c** Loading scores of the genes for PC1 during the myogenesis of pgEpiSCs in vitro. **d** A heatmap of clusters representative of each cell population during the myogenesis of pgEpiSCs in vitro shows similar expression patterns of genes in the same cluster. **e** Enriched gene ontology (GO) terms in representative clusters that had high *q*

values in different cell populations during myogenic differentiation of pgEpiSCs. **f**, **g** Expression of genes related to myofibre maturation and collagen formation during myogenic differentiation of pgEpiSCs as evaluated by qPCR. WT undifferentiated pgEpiSCs, Myo-Dif terminally differentiated cells after N2 treatment. For (**a**, **b**, **d**), pgEpiSCs-MPCs: pgEpiSCs-derived myogenic progenitor cells; pgEpiSCs-MCs: mature muscle fiber cells after N2 treatment. For (**f**, **g**), error bars indicate means ± SD, *n* = 3. *** *p* < 0.001, **** *p* < 0.0001. Similar results were obtained in three independent experiments and represent significant using two-tailed student's *t* test. Exact *P* values are listed in Source Data Fig. 3.

enriched at the corresponding phase (Supplementary Fig. 7e), supporting the accuracy of the K-means clustering. Each cluster displayed distinct transcriptional alterations, with seven clusters matching the expression patterns of characteristic genes during myogenesis (Fig. 3d), Clusters 1 and 7 having PSC characteristics (Supplementary Fig. 8a, b), Clusters 2 and 3 having MPC characteristics (Supplementary Fig. 8c, d), and Clusters 5, 8, and 10 having mature myoblast and myofiber characteristics (Supplementary Fig. 8e–h). Highlighting myogenesis patterns generally ranged from pgEpiSCs to multi-nucleated myotubes by GO term functional enrichment analysis (Fig. 3e), and remodeling of the extracellular matrix was observed, which was consistent with the qPCR results (Fig. 3f, g). Furthermore, Venn diagrams revealed that both MCs from pMuSCs and pgEpiSCs were enriched for terms like muscle development-related functions and showed similar gene expression associated with myogenic development during differentiation (Supplementary Fig. 7f, g), implying that SMFs derived from pgEpiSCs are similar to those derived from pMuSCs. In general, the results indicated that pgEpiSCs-derived skeletal MCs exhibit typical muscle cell transcriptome features.

### Metabolite analysis of the muscle cells generated from pgEpiSCs

The cell differentiation process is accompanied by changes not only in gene expression but also in energy metabolism, enabling stem cells to satisfy the requirements of lineage specification[39]. To investigate the changes in metabolites during myogenic differentiation of pgEpiSCs, we performed an untargeted metabolomics analysis (Fig. 4a).

The quality control was performed first to confirm the accuracy of the metabolomic sequencing data (Supplementary Fig. 9a, b) and to observe that the two groups (pgEpiSCs and pgEpiSCs-MCs) were distinguishable (Fig. 4b and Supplementary Fig. 9c–e) when analyzed with three different methods (PCA, PLSDA, and OPLS-DA), implying that the data were reliable. We identified a total of 70 different metabolites (Fig. 4c) and performed hierarchical clustering of all significant metabolites (Fig. 4d) to better understand the differences in metabolite abundance among different samples. Differentially abundant metabolites were shown to be enriched in multiple distinct metabolic pathways (Fig. 4e), including vitamin B6, the pentose phosphate pathway, mineral absorption, protein digestion and absorption, which were identified as myogenic differentiation stage alterations of crucial pathways[40]. Moreover, succinic acid expression was down-regulated, while glucose 6-phosphate expression was up-regulated (Fig. 4g). Furthermore, the genes associated with the production of acetyl-coenzyme A, ATP, and glycolysis metabolism-related enzymes exhibited significant changes after myogenic differentiation, as confirmed by qPCR (Fig. 4f, h). In addition, compared to undifferentiated cells, differentiated cells produced significantly more ATP and lactate but less glucose (Fig. 4i and Supplementary Fig. 9f, g). In conclusion, these results established the metabolomic characteristics of pgEpiSCs-derived skeletal MCs, validating the feasibility of differentiation at the metabolic level.

### Plant-based 3D edible scaffolds enable the creation of meat-like tissue

Innovations in scaffold technology that support cell growth and morphology could provide textural properties to CM. We designed plant-based 3D edible scaffolds without animal-derived components by achieving ionic cross-linking between konjac glucomannan (KGM), sodium alginate (SA), and calcium ions through the substitution of an ethanol-water solution (Supplementary Fig. 10a). Considering the physico-mechanical characteristics and basic suitability of scaffolds with various compounding ratios for MCs culturing, we compared the structural characteristics (Supplementary Fig. 10b, c, f, g), the inoculation efficiency, adhesion, and survival of C2C12 and pMuSCs (Supplementary Fig. 10d, e, h), water absorption and degradation (Supplementary Fig. 11a–c), and mechanical stability (Supplementary

Fig. 11d) of scaffolds. The results showed that the $Ca^{2+}$-$KGM_5$-$SA_5$ scaffold not only has a porous and homogeneous lamellar structure but also has high water uptake and low degradation properties. It also exhibits over 95% adherence to live cells and stable mechanical support. Thus, the $Ca^{2+}$-$KGM_5$-$SA_5$ scaffold could be used as the scaffold type for future research.

To acquire CM derived from pgEpiSCs, we assessed the ability for initial culture, proliferation, and myogenic differentiation by inoculating pgEpiSCs-myoblast on the $Ca^{2+}$-$KGM_5$-$SA_5$ scaffold (Fig. 5a). Cells adhered to the scaffolds with a high survival rate (Fig. 5b and Supplementary Fig. 12b, c) and significant proliferation (Supplementary Fig. 12a). Further study revealed that pgEpiSCs-MCs exhibited interconnected, extended myofiber morphology, covered the porous structure (Fig. 5c, d and Supplementary Fig. 12f), and expressed myofiber differentiation-related proteins (Supplementary Fig. 12h, i). Besides, pgEpiSCs-MCs showed high mRNA expression of genes related to maturation of myogenic differentiation (*MYOG*, *MYMK*), skeletal muscle fibers (*MYH2*, *MYH3*), and collagen formation (*COL3A1*, *COL5A2*) on 3D scaffolds, while the pluripotency genes (*OCT4*, *NANOG*) were down-regulated (Fig. 5e).

Eventually, CMs produced from pgEpiSCs stained with food coloring had a meat-like appearance (Fig. 5f and Supplementary Fig. 12j) and enhanced textural properties compared to scaffolds of uninoculated cells (Fig. 5g), especially after cooking (Supplementary Fig. 12k). Importantly, CMs produced by pgEpiSCs were rich in conditional essential amino acids and free of contamination by foodborne pathogenic bacteria, while inoculated cells maintained stable karyotypes (Supplementary Fig. 12d, e, g). The total protein content in pgEpiSCs-CM was around 3.92% compared to the plant-based scaffolds without cell inoculation, and trace elements (such as Zn, Ca, Mg, and K) were found (Supplementary Table 2). Overall, we have achieved the initial generation of pgEpiSCs-derived CM.

## Discussion

CM, a cutting-edge technology in cellular agriculture[2,15,19,21], aims to differentiate stem cells into mature muscle fibers in vitro by using tissue engineering techniques and is expected to address the imbalance between population growth and meat demand[1,2]. However, obtaining initiating cell lines, establishing serum-free culture systems, and constructing 3D edible scaffolds remain challenges. In this study, we achieved myogenic differentiation of stably passaged pgEpiSCs in vitro, not only employing serum-free conditions throughout the process but also achieving the generation of CM from pgEpiSCs on a screened 3D edible scaffold of non-animal origin.

One of the challenges in the development of CM is the availability of long-term stable expansion of cell lines in vitro[3,19]. Previous studies have used MuSCs[41–44] of various species for CM development; however, these cells can not be expanded stably in vitro[25,31]. Recently, immortalized fibroblasts or bovine MuSCs as initiating cells have been demonstrated to generate the required number of cells, but the regulating mechanism remains to be elucidated[26,27]. Although induced pluripotent stem cells have been generated from various livestock animals[45,46], their application in CM development has raised safety concerns due to the requirement for exogenous gene introduction or the use of viruses during the induction process[19]. The use of PSCs does not entail the problems mentioned above, as they can self-renew indefinitely and differentiate into all cell lineages[29]. Our results further indicated that pgEpiSCs have the molecular characteristics of pluripotency and differentiation potential independent of cell generation expansion. Moreover, pgEpiSCs possess genomic stability, and we confirmed that the high-generation cells did not undergo significant DNA damage or exhibit cancer cell-related characteristics[32,33] at P200. We are continuing these passaging experiments, and the results support the great potential of pgEpiSCs for biosafety and promoting cellular agriculture applications[28].

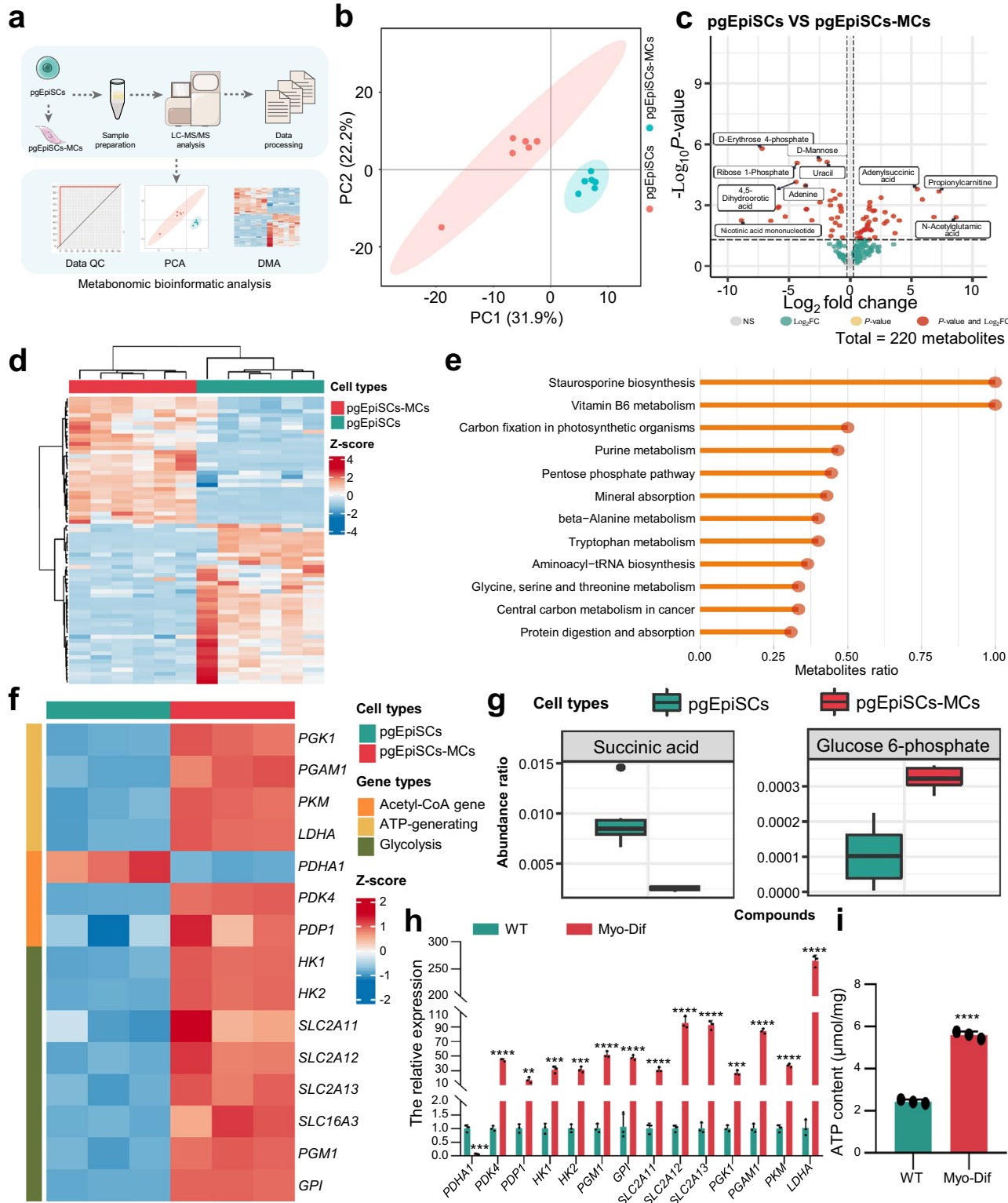

A chemically defined serum-free proliferation and differentiation system is essential in the production of CM[19,25]. Previous research has focused on transgenic[45] and serum-dependent[31] approaches for myogenic differentiation of PSCs or MuSCs, and recently, a serum-free proliferation and differentiation system was used for bovine MuSCs[30]. Although human or mouse PSC lines and myogenic differentiation systems have been established[35,47], extending the myogenic differentiation of PSCs to livestock species is a critical challenge[19]. In this study, we optimized the differentiation system to complete the entire process from pgEpiSCs proliferation to myogenic differentiation without transgenes or serum, which is expected to increase the reliability of CM development. And the accuracy of the differentiation system at the transcriptional and metabolic levels was further verified[40,48–52]. Although a clear mechanism of action for these pathways has not yet been described, these data help to understand the metabolic behavior of MCs during serum-free induction and offer a possible technical approach to further optimize media formulations by controlling the metabolic level[30].

**Fig. 4 | Metabolomic analysis of pgEpiSCs and pgEpiSCs-MCs. a** Graphic illustration of the workflow for acquiring metabolome data from pgEpiSCs undergoing myogenic differentiation. QC quality control, DMA Differential metabolites analysis. **b** PCA of pgEpiSCs and pgEpiSCs-MCs metabolome samples. **c** Volcano plot of differentially abundant metabolites between pgEpiSCs and pgEpiSCs-MCs. **d** Heatmap of differentially abundant metabolites in pgEpiSCs and pgEpiSCs-MCs. The color (blue to red) indicates metabolite abundance from low to high. **e** Kyoto Encyclopedia of Genes and Genomes (KEGG) pathway terms are enriched among metabolites of myogenic differentiation between pgEpiSCs and pgEpiSCs-MCs. **f** RNA-Seq data showing the expression of genes involved in acetyl-CoA generation, ATP generation, and glycolysis in pgEpiSCs and pgEpiSCs-MCs. **g** Box plot of differentially abundant metabolites (succinic acid, glucose 6-phosphate) in pgEpiSCs

and pgEpiSCs-MCs. The center line denotes the median, while the box contains the 25th to 75th percentiles and the whiskers mark 1.5x interquartile range. **h** qRT–PCR analysis of mRNA levels of genes associated with acetyl-CoA production, ATP production, and glycolysis. **i** Total ATP levels in pgEpiSCs and pgEpiSCs-MCs. The ATP concentration was converted to µmol/mg of protein to eliminate errors due to differences in protein amounts in sample preparation. For (**h**, **i**), WT: undifferentiated pgEpiSCs; Myo-Dif: mature muscle fiber cells after N2 treatment. Error bars indicate means ± SD, $n = 3$. ** $p < 0.01$, *** $p < 0.001$, **** $p < 0.0001$. Represent significant using two-tailed student's $t$ test and similar results were obtained in three independent experiments. Exact $P$ values are listed in Source Data Fig. 4.

Scaffolds entirely free of animal products are another key technology driving CM development, which needs to overcome the challenge of cell adhesion[43,53]. Previous studies focused on plant-based scaffolding materials[41,54] (e.g., texturized soy protein[27,44], wheat gluten[41], peanut wire-drawing protein[55,56], prolamins[57] or decellularized plants[58]), which still require small amounts of animal-derived components[43] (e.g., fibrinogen[59], thrombin[44], collagen[60,61] and gelatin[42,43,49,62]) during inoculation to promote adhesion. The studies based on plant polysaccharides also provide ideas to reduce the addition of animal-derived components[54], but the chemical modifications are difficult to achieve in terms of safety and cost control. In this study, the plant-based 3D edible scaffold was generated with lyophilization-independent ionically cross-linked porous lamellar structures by ethanol-water solution substitution to construct pgEpiSCs-derived CMs. The scaffolds consist of plant-based polysaccharide components that can provide cell adhesion[41,63] and well-growth for pgEpiSCs-MCs without additional animal-derived adhesion materials or complex polysaccharide modifications, which not only exhibit significant cell proliferation and differentiation properties but are also rich in amino acids and free of contamination by foodborne pathogens with better safety. Moreover, the use of plant-based materials has enabled controlled costs for large-scale scaffolding production[64]. Along with its low cost, this scaffold exhibits great potential for future mass manufacturing, but its structural and mechanical features must be enhanced for future mixed cultures of different cells in addition to SMFs.

The fact that pgEpiSCs keep proliferating during a longer period of differentiation indicates that they can generate a larger cell number, but the myogenic differentiation process of pgEpiSCs is more complex than that of MuSCs or immortalized myoblast cell lines (such as bovine satellite cells[26], chicken fibroblasts[27] or C2C12), which needs to be simplified or standardized to meet the requirements for the production process of CM in the future. Besides, the problem of cost must be resolved, with the creation of chicken breast production costing between \$5.10 and \$8.20 per kilogram[27]. However, the commercial stem-cell grade serum substitutes and growth factors used in the culture medium are protected by patents and are not yet cost-controlled. Future research and development are required for the cost reduction of the culture medium, such as the later large-scale production of growth factors by biosynthesis and serum substitutes according to food industrial product standards, which are also the projects we are currently tackling. Moreover, the technique of large-scale cell suspension culture is projected to overcome the limit of cell quantity and thereby satisfy the demands of industrial applications.

In conclusion, we have generated pgEpiSCs-derived CM that could overcome challenges related to cell expansion and serum dependence. This work is expected to enhance the utilization of PSCs in the CM industry, not only providing a technological approach for the development of CM but also benefitting the development of cellular agriculture and sustainable animal husbandry.

## Methods
### Animal treatment and ethics statements
All the mouse and pig experiments performed were approved by the Institutional Animal Care and Use Committee of China Agricultural University (AW 03801202-3-31).

### Mice
The Kunming white mice used in the experiments were purchased from Beijing SiPeiFu Biotechnology Co., Ltd (Beijing, China) and used for the isolation of mouse embryonic fibroblasts (MEFs) for pgEpiSCs culture. All mice were individually housed under a 12 h light/dark cycle in a sterile environment and provided with food and water ad libitum.

### Pigs
The Nongda Xiang pigs (1-week-old) were obtained from the China Agricultural University Experimental Miniature Pig Farm, and used for the isolation of porcine MuSCs.

### The isolation of MEFs
The mouse fetuses (E13.5) were obtained from the pregnant mouse's uterus and placed in a 10 cm dish containing DPBS (Gibco, Cat# C14190500CP) with 1% penicillin-streptomycin (Thermo Fisher Scientific, Cat# 15140-122). The torso (without head, limbs, tail, and viscera) of the fetuses were minced by using sterilized surgical scissors and forceps in 1.5 mL centrifuge tubes, which were digested by using trypsin-EDTA (Gibco, Cat# 25300120) in a water bath for 5–10 min at 37 °C. The cell suspension was moved to a 10 cm plate and cultured at 37 °C in an incubator with 5% $CO_2$ after the digestion was stopped using MEFs medium (DMEM (Gibco, Cat# 11960-044) supplemented with 10% FBS (Gibco, Cat# 16000-044) and 1% penicillin-streptomycin (Thermo Fisher Scientific, Cat# 15140-122)). The MEFs used for PSC culture (named feeders) need to be treated with mitomycin C (13.3 µg/mL, Selleckchem, Cat# S8146) for 3.5 h and washed three times with DPBS (Gibco, Cat# C14190500CP). Cells were digested for 5 min using trypsin-EDTA (Gibco, Cat# 25300120), and an equal volume of medium was added for neutralization. They were centrifuged for 5 min at $1500 \times g$ to harvest the cellular precipitates. Cells can be frozen at a density of $2.4 \times 10^6$ cells/mL, which can be thawed and inoculated into culture wells before inoculation with pgEpiSCs.

### Culturing pgEpiSCs
The culture medium (pgEpiSCs culture medium in Supplementary Table 1) was configured as previously[29] described, pgEpiSCs were maintained on the MEF feeders, which were washed with DPBS (Gibco, Cat# C14190500CP) and treated by Acctuase (Gibco, Cat# A11105-01) for 5 min to passage each 2–3 days at a density of $3$–$5 \times 10^4$ cells/cm². 

### Myogenic differentiation of pgEpiSCs by serum-free induction
The procedure of myogenic differentiation was adapted from a previously published study[35] and modified to allow for myogenic differentiation of pgEpiSCs. The pgEpiSCs were digested into single cells and

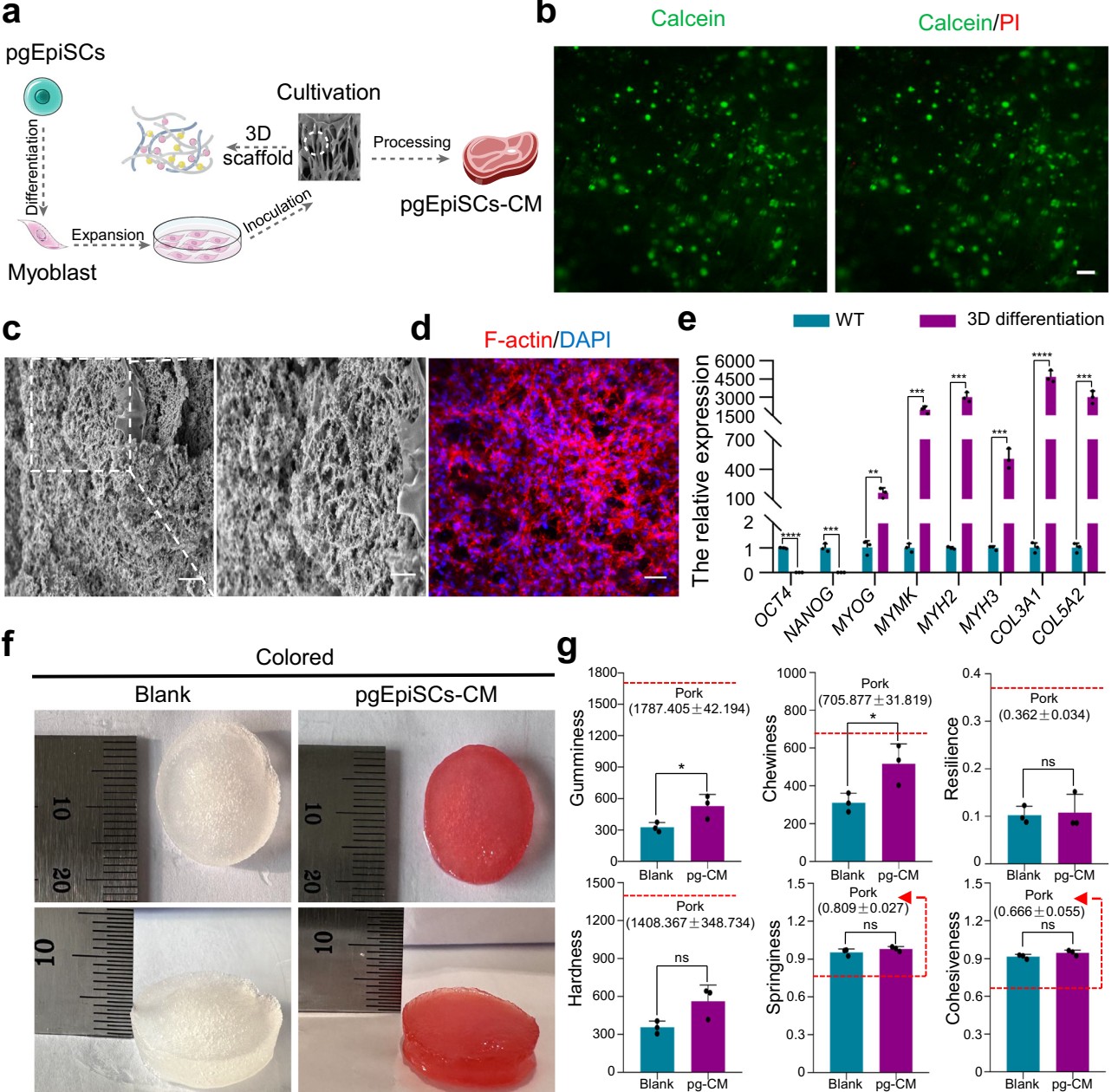

**Fig. 5 | pgEpiSCs-derived myoblast attachment and maturation on plant-based 3D scaffolds of Ca²⁺-KGM₅-SA₅. a** Schematic diagram of the inoculation of pgE-piSCs differentiated to stage V and subsequently inoculated on plant-based 3D edible scaffolds, proliferated for 5 days, and cultured in N2 serum-free terminal differentiation medium for 5 days. pgEpiSCs-CM: pgEpiSCs-derived CM. **b** The cells were stained with Calcein-AM (green) and PI (red) after inoculation into the scaffold culture for 24 h. Scale bar, 100 μm. **c** SEM images of pgEpiSCs-derived CM after culture. Scale bar, 100 μm (left) and 50 μm (right). **d** Confocal microscopy to observe the differentiation status on plant-based 3D edible scaffolds, representative fluorescent images of cytoskeletal protein F-actin in day 10. Red, phalloidin; blue, DAPI. Scale bar, 50 μm. **e** Quantification of mRNA expression of pluripotency

(*OCT4*, *NANOG*), maturation of myogenic differentiation (*MYOG*, *MYMK*), skeletal muscle fibers (*MYH2*, *MYH3*) and collagen formation (*COL3A1*, *COL5A2*) related genes by qRT-PCR. WT: undifferentiated pgEpiSCs; 3D differentiation: pgEpiSCs-MCs differentiation on 3D scaffolds. **f** Appearance of stained pgEpiSCs-derived CM in culture for 10d. **g** Textural properties of pgEpiSCs-derived CM in culture for 10d. The red line indicates the value of the fresh pork. Blank: empty scaffolds without cell inoculation; pg-CM: pgEpiSCs-derived CM. For (**e**, **g**), error bars indicate means ± SD, n = 3. * $p < 0.05$, ** $p < 0.01$, *** $p < 0.001$, **** $p < 0.0001$, and ns indicates $p > 0.05$. Represent significant using two-tailed student's *t* test and similar results were obtained in three independent experiments. Exact *P* values are listed in Source Data Fig. 5.

harvested using Acctuase (Gibco, Cat# A11105-01), which were rested in the incubator for 30 min to remove the feeder and seeded on gelatin-coated (Stem Cell Technologies, Cat# 07903) plates at a density of $1.5 \times 10^5$ cells/cm² to induce myogenic differentiation. All pgE-piSCs differentiated cells were dissociated with TryPLE (Gibco, Cat# 12605010) and harvested cells were needed to be supplemented with 10 μM Y27632 (Selleckchem, Cat# S1049) for 24 h following

inoculation, which with new medium replaced every other day. The approach for myogenic differentiation was divided into the following major steps, and the culture medium configuration is shown in Supplementary Table 1.

1. Adherent medium optimization. The culture of basic medium (BM) configuration was optimized based on the published literature. For details, see adherent medium in Supplementary Table 1.

2. Optimization of myogenic differentiation medium (MDM) induction approaches. For details, see MDM in Supplementary Table 1. (I) Single cells of pgEpiSCs were resuspended in MDM I and maintained for 3 days, which was optimized from previous studies[34–37]: (a) MDM BM supplemented with 1% B27 (Thermo Fisher Scientific, Cat# 12587-010), 3 μM CHIR99021 (Selleckchem, Cat# S1263) and 0.5 μM LDN193189 (Stemgent, Cat# 04-0074); (b) MDM BM supplemented with 1% B27 (Thermo Fisher Scientific, Cat# 12587-010), 3 μM CHIR99021 (Selleckchem, Cat# S1263) and 2 μM SB431542 (Selleckchem, Cat# S1067); (c) MDM BM supplemented with 1% B27 (Thermo Fisher Scientific, Cat# 12587-010), 3 μM CHIR99021 (Selleckchem, Cat# S1263), 0.5 μM LDN193189 (Stemgent, Cat# 04-0074) and 2 μM SB431542 (Selleckchem, Cat# S1067). (II) Cells were harvested using TryPLE (Gibco, Cat# 12605010) after 3 days of induction and inoculated in MDM II. (III) The culture was replaced with MDM III for 2 days. (IV) After 2 days of induction, MDM IV was supplied and maintained for 4 days. (V) Incubate for 20-25 days in MDM V. (VI) Replaced V with N2 (Thermo Fisher Scientific, Cat# 17502-048) differentiation medium.

## Isolation and culture of porcine MuSCs

Porcine MuSCs were isolated from 1-week-old pigs. 1 g of the longest dorsal muscle was washed in DPBS (Gibco, Cat# C14190500CP) containing 10% penicillin-streptomycin (Thermo Fisher Scientific, Cat# 15140-122), which was minced with surgical scissors and added to the digestion working solution (each 10 mL of digestion working solution contained 6.9 mL DMEM (Gibco, Cat# 11960-044), 1.5 mL of collagenase II (Coolaber, Cat# CC3791G) stock solution (10×), 1.5 mL of Dispase II (Coolaber, Cat# CD4691) stock solution (10×), and 1% penicillin-streptomycin (Thermo Fisher Scientific, Cat# 15140-122)), shaking digestion at 37 °C for 1 h. After terminating the digestion with complete culture medium (DMEM/F12 (Thermo Fisher Scientific, Cat# 10565-018) supplemented with 1% penicillin-streptomycin (Thermo Fisher Scientific, Cat# 15140-122), 1% MEM Non-Essential Amino Acids Solution (Thermo Fisher Scientific, Cat# 1140-050), 10% FBS (Gibco, Cat# 16000-044), and 5 ng/mL FGF$_2$ (PeproTech, Cat# 100-18B)) and screening successively. The cell suspensions were centrifuged at 1500 × $g$ for 5 min and resuspended in Red Blood Cell Lysis Buffer (Solarbio, Cat# R1010), placed on ice for an additional 5 min, and then centrifuged at 450 × $g$ for 10 min at 4 °C. The cell precipitation was resuspended with complete culture medium, inoculated onto T25 culture dishes, and cultured at 37 °C in a 5% CO$_2$ incubator. The differential apposition method was used to purify MuSCs, which involved aspirating the supernatant after 2 h of growth and transferring it to a fresh cell culture dish. Porcine MuSCs were cultured on gelatin-coated (Stem Cell Technologies, Cat# 07903) cell culture plates with complete medium and the media was replaced every two or three days, while cell fusion reached 90%, the cultured medium was changed to MuSCs differentiation medium (DMEM (Gibco, Cat# 11960-044) supplemented with 1% penicillin-streptomycin (Thermo Fisher Scientific, Cat# 15140-122) and 2% HS (Gibco, Cat# 26050088)) for 4 days.

## Embryoid body (EB) differentiation

Different generations of pgEpiSCs were grown to ~85% fusion, digested by Accutase (Gibco, Cat# A11105-01) for 5 min to dissociate into single cells, and inoculated at a density of 1.0 × 10$^5$ cells on 3.5 cm low-attachment dishes with differentiation medium (DMEM (Gibco, Cat# 11960-044) supplemented with 10% FBS (Gibco, Cat# 16000-044), 1% penicillin-streptomycin (Thermo Fisher Scientific, Cat# 15140-122), 1% GlutaMAX (Thermo Fisher Scientific, Cat# 35050-061)) for 7 days at 37 °C, 5% CO$_2$, 75 × $g$. EBs with excellent spheroid shape were chosen and inoculated on gelatin-pre-incubated plates for 10 days before collecting or fixing the matching cell samples for future tests.

## Alkaline phosphatase (AP) staining

The AP staining assay kit (Millipore, Cat# SCR004) was performed exactly as recommended. The pgEpiSCs and its differentiated cells were fixed with 4% PFA for 5 min, which were washed with DPBS (Gibco, Cat# C14190500CP) and incubated with AP staining working solution (A:B:DPBS = 2:1:1) for 15–20 min at 37 °C in a 5% CO$_2$ incubator sheltered from light. DPBS washing was performed for observation.

## Karyotype analyses

The proliferating pgEpiSCs were incubated in culture medium supplemented with 15% colcemid solution (Gibco, Cat# 15210-040) for 2.5 h. Cells were harvested and resuspended in 10 mL of 75 mM KCl solution and incubated at 37 °C for 30 min with blowing or turning every 5 min. Followed by addition of pre-cooled fixative solution (methanol/glacial acetic acid 3:1), centrifugation at 2000 × $g$ for 5 min, resuspension of precipitate in pre-cooled fixative solution and incubation on ice for 30 min, repeated once and incubation on ice for 1 h. Cells were dropped onto glass slides, which were dried in an oven at 37 °C and stained for visualization using a Rapid Giemsa Staining kit (BBI Life Science, Cat# E6073141).

## Alkaline comet assay

We used an alkaline comet assay experiment to determine the degree of DNA damage at the single-cell level, which is known as the single-cell gel electrophoresis. It was performed with reference to the description of the previous studies[65] and with a slight modification of the method. Accutase (Gibco, Cat# A11105-01) was used to separate different generations of pgEpiSCs into single cells, which were then resuspended in ice-cold DPBS (Gibco, Cat# C14190500CP) with the cell density adjusted to 1.0 × 10$^6$ cells/mL. A suspension of 30 μL cells was mixed thoroughly with 70 μL of 0.7% low-melting agarose, and the cell-agarose suspension was applied fast and evenly dropwise onto comet slides, which were kept at 4 °C for 10 min followed by subsequent immersion in lysis buffer (2.5 M NaCl, 100 mM Na$_2$EDTA, 10 mM Tris, 1% N-lauroylsarcosine, 1% Triton X-100, pH 10.0) for 1 h protected from light. The slides were washed with DPBS (Gibco, Cat# C14190500CP) for 3 min each time before being submerged in cold electrophoresis buffer (300 mM NaOH, 1 mM EDTA, pH > 13) for 30 min at 4 °C, which shielded them from light with electrophoresis for 30 min (1 V/cm, 300 mA, 4 °C). The slides were submerged in neutralization buffer (0.4 M Tris-HCl, pH 7.4) for 5 min after electrophoresis before being maintained in cold 100% ethanol for 1 h. After drying at room temperature away from light, the slides were stained for 30 min with nucleic acid dye (GenStar, Cat# ZE111-101S) and photographed for recording. CASP Comet assay software was used to evaluate the samples, and at least 50 cells were counted for each sample. Experiments were performed with 200 μM H$_2$O$_2$-treated (Sigma, Cat# 316989) pgEpiSCs as a positive control group.

## Immunofluorescent staining

Cells were fixed for 30 min in freshly prepared 4% PFA at room temperature, permeabilized for 20 min in 0.5% Triton X-100, and blocked with 3% BSA (Sigma, Cat# A1470) for 2 h at 4 °C. Primary antibodies (Mouse-OCT3/4 (Santa Cruz Biotechnology, Cat# sc-5279), Mouse-SOX2 (Santa Cruz Biotechnology, Cat# sc-365823), Rabbit-NANOG (PeproTech, Cat# 500-P236), Rabbit-tubulin (Abcam, Cat# ab18207), Rabbit-SMA (Abcam, Cat# ab5694), Rabbit-Vimentin (Abcam, Cat# ab92547), Rabbit-Brachyury (Santa Cruz Biotechnology, Cat# sc17743), Mouse-PAX7 (DSHB, Cat# PAX7-S), Rabbit-MYOD (Proteintch, Cat# 18943-1-AP), Mouse-Myosin (Sigma, Cat# M4276), Mouse-MyHc (DSHB, Cat# MF20-S)) were incubated on cells at 4 °C overnight, followed by incubation with diluted secondary antibodies for 1 h. The Actin-Tracker Red (Beyotime, Cat# C2205S) was incubated on cells at 4 °C for 2 h. Cells were washed three times in DPBS (Gibco, Cat# C14190500CP) and stained with DAPI (Roche Life Science, Cat#

10236276001) for 5 min, which were photographed under a fluorescence microscope. The fusion index was calculated as the percentage of nuclei in a single fused muscle fiber out of the total nuclei. The information on antibodies is listed in Supplementary Table 1.

## Flow cytometric analysis

Cells were digested by TryPLE (Gibco, Cat# 12605010) for 5 min into single cells and centrifuged at 1500 × g for 5 min, which were washed three times with DPBS (Gibco, Cat# C14190500CP) before being resuspended with diluted antibodies (Alexa Fluor 647 anti-pig CD45 (BIO-RAD, Cat# MCA1222A647), APC-conjugated anti-pig CD31 (BIO-RAD, Cat# MCA1746APC), PE-conjugated anti-human CD56 (BioLegend, Cat# 304606)) and incubated on ice for 30 min. Followed by washing three times with cold DPBS (Gibco, Cat# C14190500CP) and resuspending precipitates for flow analysis of CD31$^-$CD45$^-$CD56$^+$ ratios, where unstained cells were used as negative controls for delineation of FACS gating parameters and the differentiation efficiency was calculated as the proportion of CD31$^-$CD45$^-$ cells multiplied by the proportion of CD56$^+$ cells. The information on antibodies is listed in Supplementary Table 1.

## Quantitative RT-PCR

Total RNA was extracted using the RNAprep pure Cell/Bacteria Kit RNAprep pure (TIANGEN, Cat# DP430), and its concentration was measured, which was then inverted to cDNA using Hifair® III 1st Strand cDNA Synthesis SuperMix (YEASEN, Cat# 11141ES60) for qPCR. RT-qPCR by using 2 × RealStar Green Power Mixture (GenStar, Cat# A311-05). Primer information for RT-qPCR was shown in Supplementary Table 1. The resulting cycle thresholds (CT) were analyzed using the comparative CT (2$^{-\Delta\Delta CT}$) approach and *EF-1α* was determined as an internal control. All experiments included three biological replicates and primer sequences are shown in Supplementary Table 1 for primers.

## Transcriptome sequencing

**Library preparation.** The RNA from the quality-checked samples as the starting sample, and the RNA with Poly-A structure in the eukaryotic total RNA was enriched with the TIANSeq mRNA capture kit (TIANGEN, Cat# NR105), and the captured RNA as the initial sample for library construction with the VAHTS Universal V6 RNA-seq Library Prep Kit for Illumina (VAZYME, Cat# NR604-02). Following the construction of the library, Qubit2.0 was used for initial quantification, and the Agilent 2100 Bioanalyzer was utilized to evaluate the insert size of the library. After the insert size was determined to be acceptable, Q-PCR was performed to precisely quantify the effective concentration of the library (effective library concentration >2 nM) to guarantee the library's quality. Furthermore, separate libraries were combined based on effective concentration and target downstream data volume, and PE150 sequencing was done on the Illumina platform to get 150 bp double-end sequencing reads.

**Raw data quality control, alignment to the genome and analysis of differential expression genes (DEGs).** For the paired-end data, trimgarole (v-0.6.6) software was used for quality control with the parameters: "-q 25 -j 8 --phred33 --length 74 -e 0.1 --stringency 4 --paired". Then, high-quality reads were mapped to the reference pig genome (Sscrofa 11.1) using HISAT2 (v-2.1.0) with default parameters[66]. Expression levels of all genes were quantified as read counts using FeatureCounts (v-2.0.1) and the parameters for all exons were calculated. We calculated the TPM in R for all samples and only considered a protein-coding gene as detected if its TPM value was greater than 0.5 in at least half replicates of all biological replicates. DEGs between different samples were identified using the DESeq2 tool with the Wald-test (v-1.30.1).

**Principal component analysis.** Analyses of all samples PCA plots were performed using the R package FactoMineR (v-2.4) and visualization

was performed using the R package Factoextra (v-1.0.7). The loading score of PC1 was calculated by the "prcomp" function.

**Correlation matrices and heatmaps.** The correlation matrices of all samples were generated using Pearson's correlation coefficient for all detected genes by the "cor" function. R package pheatmap (v-1.0.12) was used to display correlation coefficient matrices and gene expression matrices.

**Volcano plots.** Volcano plots were plotted by R packages EnhancedVolcano (v-1.14.0), with a cutoff of $\log_2^{FoldChange} = 2$, padj = 0.05.

**Ternary plots.** Ternary plots were produced with the R package ggtern (v-3.3.5) using the average expression for each chosen group. Key genes of each cell type were highlighted. Density areas were computed using 2D kernel density estimation.

**Construction of expression tendencies.** To systematically explore the characteristics of pgEpiSCs during myogenic differentiation, we constructed the expression tendencies by median TPM value in the selected samples. We first calculated the median TPM value for each cell type separately. The median TPM value was rescaled and analyzed by the k-means clustering method with parameters $k = 10$ and iter.-max = 100. The average and standard deviation of the median expression levels of the logarithm for each cluster were calculated to evaluate the performance of clustering. We then selected clusters representing the respective characteristics of each cell type for further analysis.

**Functional enrichment analysis.** Functional enrichment analysis of selected genes was performed using Metascape[67] (http://metascape. org). Genes were mapped to their respective human orthologs, and the lists were submitted to Metascape for enrichment analysis based on the significant overrepresentation of GO biological processes (GO-BP) categories. Only GO-BP terms with resulting minimum count genes ≥3 and adjusted $p < 0.05$ were considered significant and were depicted using the lollipop plot in ggplot2.

## Untargeted metabolomic determination and analysis

**Sample preparation.** The harvested cell samples were placed in a liquid nitrogen tank with 400 μL of extraction solution (methanol:acetonitrile:water = 2:2:1, containing an isotope-labeled internal standard mixture). They were then sonicated for 10 min in an ice water bath before resting for 1 h at −40 °C. After centrifuging the samples at 4 °C for 15 min at 13,400 × g, the supernatant was recovered from the injection vial and analyzed on the machine. After that, all samples were mixed into QC samples with comparable amounts of supernatant.

**Measurement of metabolites.** The target compounds were separated using a Vanquish (Thermo Fisher Scientific) Ultra Performance liquid chromatograph on a Waters ACQUITY UPLC BEH Amide liquid chromatographic column. The aqueous A phase of fluid chromatography included 25 mmol/L ammonium acetate, 25 mmol/L ammonia, and the acetonitrile B phase. The temperature of the sample tray was 4 °C, and the injection volume was 2 μL.

**Metabolomics data processing and analysis.** All LC/MS acquired raw data were converted to mzXML using ProteoWizard (v-3.0.22248). We then perform peak identification, peak extraction, peak alignment, and integration using R package XCMS (v-3.10.2). Subsequently, the metabolites were identified by matching them with compounds in the mzCloud database, and then selected compounds with CV (coefficient of variance) values less than 30% in QC pool samples as identification results for subsequent analysis. Using Compound Discover software (v-3.0), the chromatographic peaks detected in the samples were

integrated, in which the peak area of each characteristic peak represented the relative quantitative value of a compound. The quantitative results were normalized using the total peak area, and finally the quantitative results of metabolites were obtained. IP4M (v-2.0) and mixOmics (v-6.14.1)[68] were used for PCA, PLSDA, OPLDSA, ROC enrichment analysis, and other data processing.

### ATP, glucose, and lactic acid measuring
Intracellular ATP levels were measured using the ATP Assay Kit (Beyotime, Cat# S0027) and rigorous manipulation as instructed by the manufacturer. Acctuase (Gibco, Cat# A11105-01) was used to dissociate pgEpiSCs into single cells, while TryPLE (Gibco, Cat# 12605010) was used to dissociate pgEpiSCs-MCs into single cells. After centrifuging at $1500 \times g$ for 5 min and removing the supernatant, 200 μL of lysis buffer was added to resuspend the cells. Cell precipitates were maintained on ice for 20 min before being centrifuged at 4 °C at $12,000 \times g$ for 5 min. The supernatant was taken for further analysis. Add 100 μL of ATP assay buffer to each well of a 96-well microtiter plate and stand for 5 min, then add 40 μL of sample and mix quickly, then measure the RLU value with a multimode reader (Tecan, Spark). The standard curve was used to calculate the concentration of ATP in the samples. Protein concentrations were used to adjust relative ATP levels. The medium's glucose (Beyotime, Cat# S0201M) and lactate (Nanjing Jiancheng Institute of Biotechnology, Cat# A019-2-1) contents were measured in strict accordance with the kit's usage instructions.

### Preparation of 3D edible scaffolds
The plant-based 3D edible scaffolds were created by freeze-drying independent porous materials by solution substitution with ionic cross-linking. Briefly, a 75% ethanol aqueous solution of 2% (w/v) CaCl₂ (Sigma-Aldrich, Cat# C5670), aqueous solutions of 3% (w/v) SA and 3% (w/v) KGM were prepared, respectively, and homogeneous mixtures were prepared in different ratios of KGM:SA (9:1, 7:3, 5:5, 3:7, 1:9), transferred to 24-well plates (0.5 g/well) and pre-frozen at −20 °C for 12 h. The scaffolds were obtained by adding CaCl₂ aqueous ethanol solution (1 mL/well) and replacing them for 24 h.

### Water absorption capacity
The swelling ratio was used to evaluate the water absorption capacity of the scaffolds. The lyophilized scaffolds were placed in DPBS (Gibco, Cat# C14190500CP) at room temperature for 6 h, 12 h, 24 h and observed for changes in weight to signify fluid uptake. Surface moisture was removed from the sample by wiping with filter paper. Using a balance to measure the weight of the scaffold in the expanded scaffold (Wss) and the dry scaffold (Wds). The swelling ratio (%) of the scaffold was calculated using the following formula:

$$Swelling\ ratio\,(\%) = \left[(Wss - Wds)/Wds\right] \times 100 \qquad (1)$$

### Degradation ratio
The scaffolds were immersed in the medium at 37 °C for 1 h, and the excess solution on the surface of the scaffolds was gently removed with filter paper and weighed as W0. After weighing, the scaffolds were placed back and filled with the medium placed under the conditions of 37 °C and 5% CO₂. Then removed on Days 3, 6, 9, 12, and 15, respectively, and weighed as Wt. The degradation ratio (%) was calculated by the following equation:

$$Degradation\ ratio\,(\%) = \left[(W0 - Wt)/W0\right] \times 100 \qquad (2)$$

### Mechanical property assay
The compression tests of the scaffolds were measured by a texture analyzer (TA. XT Plus, Stable Micro Systems Ltd., UK). Briefly, the scaffolds were compressed to 75% of their initial height at 5 mm s⁻¹ with a cylindrical probe of 36 mm diameter. The compression force was recorded during the entire compression process.

### SEM analysis
The sample preparation for SEM was divided into two parts: (1) pretreatment of the lyophilized scaffolds, which were placed in liquid nitrogen to brittle the surface, cross-sections, and longitudinal sections to the proper size; (2) pretreatment of the scaffolds inoculated and cultured with pgEpiSCs-MCs, which were fixed with 4% PFA for 1 h, washed with HBSS (Beyotime, Cat# C0219), and then dehydrated using a gradual concentration of ethanol in water (30%, 50%, 70%, 80%, 90%, 100%), followed by supercritical drying. The pretreated samples were sputter-coated with Pt/Pd on SEM stubs and scanned at an acceleration voltage of 5 kV, and images were analyzed by ImageJ software to calculate pore area and pore diameter.

### Three-dimensional (3D) scaffolds for cell culturing
The scaffolds were sterilized with ionizing irradiation at 1 kGy prior. C2C12 (CRL-1772), pMuSCs, or pgEpiSCs-MCs were resuspended in media and seeded at a density of $1.0 \times 10^6$ cells/mL into 3D edible scaffolds, which were allowed to adhere for 4 h and supplied with the matching medium. C2C12 were maintained in proliferation medium (DMEM (Gibco, Cat# 11960-044) supplemented with 10% FBS (Gibco, Cat# 16000-044) and 1% penicillin-streptomycin (Thermo Fisher Scientific, Cat# 15140-122)). pMuSCs were maintained in proliferation medium (DMEM (Gibco, Cat# 11960-044) supplemented with 10% FBS (Gibco, Cat# 16000-044), 1% penicillin-streptomycin (Thermo Fisher Scientific, Cat# 15140-122), 1% MEM Non-Essential Amino Acids Solution (Thermo Fisher Scientific, Cat# 1140-050), 1% GlutaMAX (Thermo Fisher Scientific, Cat# 35050-061), and 5 ng/mL FGF₂ (PeproTech, Cat# 100-18B)). The pgEpiSCs-MCs were maintained in Stage V medium for the first 5 days and in differentiation medium containing N2 (Thermo Fisher Scientific, Cat# 17502-048) for the next 5 days. All three-dimensional cell cultures were cultured in a 37 °C, 5% CO₂ incubator with individual liquid changes.

### Assay of cell survival efficiency on 3D edible scaffolds
C2C12, MuSCs, or pgEpiSCs-MCs were cultivated on scaffolds, and cell survival status was measured using the Calcein/PI Cell Activity and Cytotoxicity Assay Kit (Beyotime, Cat# C2015M). Cells were stained with Calcein-AM (AM) and propidium iodide (PI) double fluorescence, observed with a fluorescence microscope (DM6 B, LECIA, Germany), and flow cytometry was used to determine cell survival effectiveness (BD FACSVerse). The scaffold with the highest survival rate was selected as the most proportional scaffold.

### Inoculation efficiency
After inoculation of C2C12 or pMuSCs onto the 3D scaffold and incubation for 24 h, the scaffold was removed from the medium, and cells adhering to the well plates (cells not attached to the scaffold) were digested with trypsin for cell counting, and inoculation efficiency (%) was calculated by the following formula:

$$Inoculation\ efficiency\,(\%) = \left(\frac{Initially\ seeded\ cells - Unattached\ cells}{Initially\ seeded\ cells}\right) \times 100$$

$$(3)$$

### Cell proliferation analysis
Tracking the proliferation of differentiated pgEpiSCs on scaffolds using a lab-owned reporter system cell line (NLS-GFP). Briefly, pgEpiSCs-MCs were inoculated into the scaffold at different times in the identical area to track changes in cell numbers.

## Confocal microscopy

The scaffolds were washed with HBSS (Beyotime, Cat# C0219) after inoculation of cells and fixed with 4% PFA at room temperature for 1 h. The method for 3D-staining was the same as for immunofluorescence staining, with Actin-Tracker Red-594 (Beyotime, Cat# C2205S) for 2 h to observe the morphology of the cytoskeleton F-actin. The nuclei were stained with DAPI (Roche Life Science, Cat# 10236276001) for 10 min and rinsed with HBSS (Beyotime, Cat# C0219) after staining. Images were taken using a laser-scanning confocal microscope.

## Texture profile analysis (TPA) of scaffolds inoculated with pgEpiSCs-MCs

TPA was measured by a texture analyzer (TA. XT Plus, Stable Micro Systems Ltd., UK). Briefly, pgEpiSCs-MCs were seeded on $Ca^{2+}$-KGM$_5$-SA$_5$ scaffolds and incubated for 10 days to obtain pgEpiSCs-CM. Blank scaffolds without inoculated cells, pgEpiSCs-CM, and porcine loin were tested for TPA before and after cooking with a double compression cycle test performed up to 75% compression of the original portion height at a compressive strain rate of 5 mm s$^{-1}$ twice.

## Microbiological detection of pgEpiSCs-derived cultured meat

To assess the sterility of pgEpiSCs-derived CM, the total number of colonies was determined separately for pgEpiSCs-derived CM and the medium of pgEpiSCs-MCs 3D culture (MDM V medium and N2 medium), with *E. coli* as a positive control.

## Staining and frying of pgEpiSCs-derived cultured meat

The pgEpiSCs-derived CM was stained at room temperature with edible pigments (0.0075% monascus colors and 0.05% beet red pigment) for 1 h. Then the stained tissues were fried in a pan with a little oil for 30 s.

## Western blotting

The cultured cells were harvested from the scaffolds, and total protein was extracted in RIPA buffer (Cell Signaling Technology, Cat# 9806) supplemented with protease inhibitor cocktail (Beyotime, Cat# P1050). The total protein content was determined by the BCA protein assay kit (Sangon Biotech, Cat# C503051) and 15 μg of total protein from different samples was electrophoresed by 8% sodium dodecyl sulfate polyacrylamide gel electrophoresis. Then they were transferred to polyvinylidene fluoride membranes and incubated with primary antibodies: OCT4 (Santa Cruz Biotechnology, Cat# sc-5279, 1:1000), MYH3 (Santa Cruz Biotechnology, Cat# sc-376157, 1:1000), and GAPDH (Cell Signaling Technology, Cat# 5174, 1:5000) at 4 °C overnight. The secondary antibodies were horseradish peroxidase (HRP)-conjugated anti-rabbit IgG (Beyotime, Cat# A0208, 1:10000), and HRP-conjugated anti-mouse IgG (Cell Signaling Technology, Cat# 7076, 1:10000) at room temperature for 2 h. The bands were exposed to an enhanced chemiluminescence solution (Thermo Fisher Scientific, Cat# 34075).

## Amino acid analysis

The amino acid composition of the cells was determined according to GB5009.124-2016. Basically, the samples were hydrolyzed in a hydrolysis tube with 10 mL of HCl (6 mol/L) at a constant temperature of 110 °C for 22 h. The hydrolysis products were transferred and diluted in a 50 mL volumetric flask with a fixed volume of 0.02 mol/L HCl. After mixing, 1 mL of the solution was transferred to an evaporation flask, dried by rotary evaporation in a water bath at 55 °C, then dissolved with ultrapure water, rotary evaporated, and the above two steps were repeated twice. After that, amino acids were dissolved in 2 mL of HCl (0.02 mol/L), filtered (0.22 μm), and measured using a fully automated amino acid analyzer (LA8080, Hitachi, Tokyo, Japan) for comparison with the standard (013-08391, Wako, Tokyo, Japan) for determination of individual amino acids.

## Statistical analysis

All values in the graphs are reported as the mean ± SD. Prism 9 was used to construct diagrams, and the student's *t*-test (two-tailed) was used to compare various groups. Date was subjected to a one-way ANOVA with Duncan's new multiple range test for analysis of significance, were used to compare the means of more than two groups. *P* values less than 0.05 were considered statistically significant.

## Reporting summary

Further information on research design is available in the Nature Portfolio Reporting Summary linked to this article.

## Data availability

The RNA-seq data generated in this study have been deposited in the Gene Expression Omnibus database under accession code "GSE223433". The metabolome data generated in this study have been deposited in the OMIX, under accession code "OMIX005128". All other data supporting the findings of this study are available within the article and its supplementary files. Any additional requests for information can be directed to, and will be fulfilled by, the corresponding authors. Source data are provided with this paper.

## Code availability

All the code used for data analysis is available at https://github.com/dfgao/Meat-like-data and https://doi.org/10.5281/zenodo.10117490[69].

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

## Acknowledgements

We thank S. Wang, X. Zhang, Y. Liu (Instrumental platform of state key laboratory of plant environmental resilience, China Agricultural University, Beijing, China) and J.Z. for the help on SEM and confocal imaging. This research was supported by the National Key R&D Program of China (2022YFD1302201 and 2016YFA0100202 to J.H.), the Future Functional Food R&D Program of China Agricultural University (SJ2021002004 to J.H.), the National Natural Science Foundation of China (31970825 and 31772601 to J.H., 31972099 and 32172143 to X.F.) and the China National Postdoctoral Program for Innovative Talents (BX20220344 to M.Z.), Beijing Natural Science Foundation (6192005 to S.C.), Plan 111 (B12008 to J.H.).

## Author contributions

J.H. and X.F. conceptualized this project and supervised the overall experiments. G.Z. performed pgEpiSCs culture in vitro and characteristics analysis of high and low generation. J.H., G.Z., and D.G. coordinated and performed bioinformatics analysis of the RNA-seq and Untargeted metabolomic determination sequencing data. G.Z., L.L., and A.M. completed the preparation, screening, cell inoculation of 3D edible scaffolds and SEM. M.Z. and J.Z. performed pgEpiSCs derivation. Y.Y., Y.W., and X.Z. performed flow cytometric analysis. G.Z., Q.Z., J.G., S.C., and T.C. performed qPCR primer design, quantitative validation, and immunofluorescence image processing. G.Z., L.L., T.W., and X.C. performed the preparation of the experimental schematic. J.H., G.Z., X.F., S.C., L.L., and D.G. performed manuscript writing, review, and editing.

## Competing interests

The authors declare no competing interests.
