## [Peer review file · Nature Communications]

REVIEWER COMMENTS

Reviewer #1 (Remarks to the Author):

This research highlights the successful development of a differentiation system using pgEpiSCs for CM production. It also emphasizes the potential benefits of CM and the significance of the findings for the future of CM development. All the biological and biomolecular characterizations were in-depth and well-executed. This work also established a 3D differentiation system to shape cultured tissue. The team achieved this by screening plant-based edible scaffolds of non-animal origin, namely sodium alginate (SA) and konjac glucomannan (KGM) composite hydrogel, through a physical ionic crosslinking process.

This research effectively addresses the first limitation of cultivated meat, which is cell line development. However, there are some minor issues that need to be addressed before this work is ready for publication. Please see the point-by-point comments below:

- Which part of the fresh pork was selected for the TPA test? Was it the loin or other parts?
- Besides ionic crosslinking, what is the reason behind choosing KGM and SA as the plant-based materials? There are other plant-based materials that also contain protein, such as soy, pea protein hydrolysate, zein, etc. Why did the authors not incorporate these protein-based materials to increase the final product's nutritional value?
- Why was the confocal experiment only performed with F-Actin and DAPI? Why were other immunofluorescence staining, such as Myogenin, Desmin, or Myosin heavy chain, not included to assess the cells cultured on the 3D scaffold?
- Based on the Calcein/PI staining, the cell morphology appears to be very round. How can the authors ensure that they were fully proliferated and differentiated on the scaffolds?
- Could the authors provide zoomed-in SEM images to observe how the cells interact with the scaffolds?
- Apart from the amino acid profile provided in the supplementary materials, will the authors be able to provide the total protein content and other nutritional values for the pgEpiSCs-CM sample?

Reviewer #2 (Remarks to the Author):

Overall this is an impressive piece of work, which spans a range of relevant topic areas in cultured meat, from cell line generation through to tissue formation. Whilst the results presented are certainly interesting, at this point in time, there are some major weaknesses in the study. The primary weakness that needs to be addressed is the demonstration of myogenic differentiation at the protein level, in both 2d and edible 3d constructs.

Please find attached some major and minor comments.

Major comments

The major weakness in this study is the lack of convincing evidence of myogenic differentiation, and a strong overreliance on bulk RNA-based data to try to make this point.

In general, the differentiation images provided are of poor quality, and are not quantified in any fashion (for example using fusion index, which is very standard in the field of myogenesis). From the brightfield images, it seems like a lot of cells are aligned, but probably not fused, although it is hard to tell. The fluorescent stainings likewise show little convincing evidence of fusion, even though there is plenty of signal. No undifferentiated controls are provided for comparison, nor is there any timecourse data showing the process of myogenesis. These data must be substantially improved with higher quality images at higher magnifications.

For the 3d tissues, there is again an absence of convincing differentiation data. The supplementary material (Fig. S5) does contain some myosin stainings that look promising in terms of myotube formation, although again these are provided without quantification or adequate controls for comparison. In fact, the tubes that were formed are surprisingly reminiscent of tube formation that is seen in endothelial cells and some mesenchymal cells. S5A likewise seems to show several multinucleated myotubes, but it is hard to assess by eye what proportion of cells are actually fused. Given that this point is fairly central to the author's claim to be producing differentiated meat-like tissue, these data need to be extended, improved and emphasized more heavily in the presentation of the article.

For both 2d and 3d, there are no robustly quantifiable protein measurements, for example by western blot for major muscle proteins, which needs to be addressed. A favourable comparison with traditional meat would not be required for publication, but certainly a strong induction relative to undifferentiated cells would be expected.

The RNAseq and metabolomics data are impressive, but I somewhat question the value of these data (particularly the extent to which they are presented, as two entire figures) when there is no real way to quantify these changes relative to a differentiated sample. The metabolomics data add very little to the central statement of the study. The "crucial pathways", mentioned on line 237 for example requires much more context to be relevant for the story.

The differentiation scheme seems unusually long, given that most myogenic differentiation protocols take place over a matter of days. Some timecourse data to investigate whether this process is really required might be interesting. At a bare minimum, the authors should discuss this point and the potential limitations that it might have for an cultured meat production process based on this system.

It is difficult to assess differentiation efficiency (percentage of cells/clones that participate in differentiation) as only level of differentiation is shown and compared to literature data, not positive controls (e.g. C2C12s)

Scaffold: Please show cell adhesion to scaffold material.

Line 175: the corresponding figure says 99.8%

Supplementary Fig 9H y-axis labels are missing
Supplementary Fig 10D,
please provide stress Pa, to provide reference to muscle tissue
Line 316. There are several studies in literature proving this wrong

Minor comments

"Currently accessible cell lines include mesenchymal stem cells and muscle stem cells (MuSCs), which can be amplified only a limited number of times before losing their capacity to differentiate."

This point in the introduction is not really the case, as immortalised adult stem cells, such as bovine satellite cells, are now available (Stout et al). Such lines are also available for avian species, such as chicken.

It is rather hard to know what to make of the karyotype analysis from the provided figure(s), given that no quantification nor labelling is provided.

The article and figures need to be thoroughly checked over for English spelling and grammar. F-actin should be spelt with a hyphen. 'Cancer hallmarkers' does not make sense as a graph title, and there are many other examples.

Some of the graph axes are rather strangely named, for example 'The damage levels of DNA'.

Reviewer #3 (Remarks to the Author):

I co-reviewed this manuscript with one of the reviewers who provided the listed reports as part of the Nature Communications initiative to facilitate training in peer review and appropriate recognition for co-reviewers.

RE: “Generation of three-dimensional meat-like tissue from stable pig epiblast stem cells” (NCOMMS-23-05852B)

Point-by-point response to reviewers’ comments:

REVIEWER COMMENTS

Reviewer #1 (Remarks to the Author):

This research highlights the successful development of a differentiation system using pgEpiSCs for CM production. It also emphasizes the potential benefits of CM and the significance of the findings for the future of CM development. All the biological and biomolecular characterizations were in-depth and well-executed. This work also established a 3D differentiation system to shape cultured tissue. The team achieved this by screening plant-based edible scaffolds of non-animal origin, namely sodium alginate (SA) and konjac glucomannan (KGM) composite hydrogel, through a physical ionic crosslinking process.

This research effectively addresses the first limitation of cultivated meat, which is cell line development. However, there are some minor issues that need to be addressed before this work is ready for publication. Please see the point-by-point comments below:

(1) Which part of the fresh pork was selected for the TPA test? Was it the loin or other parts?

R: Thanks for your comment. It is the loin. The porcine loin has an important economic value as an important edible part of the pig, and porcine muscle stem cells (pMuSCs) were isolated from the longest dorsal muscle, which contains porcine loin, so we chose porcine loin for comparison of textural properties with pgEpiSC-derived CM, which is described in the methods of the revised manuscript (as shown in **Line 726**).

(2) Besides ionic crosslinking, what is the reason behind choosing KGM and SA as the plant-based materials? There are other plant-based materials that also contain protein, such as soy, pea protein hydrolysate, zein, etc. Why did the authors not

incorporate these protein-based materials to increase the final product's nutritional value?

R: Thanks for your comment and question. Konjac glucomannan (KGM) and sodium alginate (SA) were chosen as plant-based materials for the following reasons: 1) As you pointed out, it was initially based on the cross-linking capabilities of SA and calcium ions¹, which are particularly advantageous for preparation process simplicity, safety, and cost control and have significant potential for large-scale creation of CM scaffolds. 2) Regarding the substance itself, SA is an anionic polymer that occurs naturally and is typically derived from brown seaweed². It has long been extensively used in the food industry due to its abundance, good biocompatibility, low cost, and mild gelation, as well as the structure of the hydrogel it creates, which is comparable to the extracellular matrix of living tissues³. 3) SA hydrogels alone are difficult to apply directly to the preparation of CM scaffolds due to the lack of mammalian cell integrin adhesion ligands⁴. Furthermore, KGM, a water-soluble polysaccharide, has drawn more attention owing to its favorable biocompatibility, biodegradability, and hydrophilicity, as well as its lack of danger and harmful effects⁵, which may be utilized as a suitable material for scaffolds made from CM. As a result, we chose to combine KGM and SA for the creation of plant-based scaffolds without animal-derived components.

We agree with the suggestion you provided, and we have also made attempts with soy protein isolate (SPI) and zein as additional scaffold production ingredients to increase the final product's nutritional value. However, the results did not match expectations (as stated below, **Response Figure 1**). We added zein to the ethanol replacement solution since it is alcohol-soluble and substituted soy protein isolate for an appropriate proportion of the original preparation solution. The results demonstrated that the addition of zein was unable to infiltrate through the scaffold and could only form a yellow zein film on the surface (**Response Figure 1A, E**). With the addition of medium, the culture system became murky, zein precipitated, and the scaffold's structure collapsed, making it difficult to carry out nutrient replenishment and unsuitable for the growth of cells (**Response Figure 1B-D, E**). Although the scaffolds were structurally intact at the initial stage of culture after the addition

of SPI replacement (**Response Figure 1A, F**), the scaffolds became soft after further inoculation with C2C12 cells and culture for 24 h (**Response Figure 1H**). It was difficult to maintain a stable structure for long-term culture, and the results of the live/dead cell staining revealed an increase in the number of dead cells with an increase in the proportion of SPI addition (**Response Figure 1G**), which requires that further new technological approaches be sought for the incorporation of plant proteins; therefore, we did not add the results in the revised manuscript.

Additionally, plant-based scaffolds (such as texturized soy protein^{6,7}, wheat glutenin⁸, peanut wire-drawing protein prolamins^{9,10}, or decellularized plants¹¹) have been reported to reduce the dependence on animal-derived components; however, they have poor cell adhesion abilities. The addition of animal-derived thrombin⁶, fibrinogen¹², collagen¹³ and gelatin¹⁴ is required to promote cell adhesion, which is also commonly used as a solution. In contrast, the advantage of our plant-based scaffolds is that the present scaffold, made utilizing KGM with SA without any animal-derived components, has a good affinity with cells and does not require the inclusion of any animal-derived components to enhance cell adhesion. Of course, we continue to strive harder to improve the nutritional value of the final product by adding nutrients like soy protein during food processing.

Response Figure 1: Preparation and feasibility analysis of Zein/SPI- Ca^{2+} -KGM-SA scaffolds. **A**, Focus diagram of the appearance of Zein/SPI Ca^{2+} -KGM-SA scaffolds, where Zein is dissolved in ethanol replacement solution according to the ratio, and SPI is the scaffold added to the KGM-SA mixing solution. **B**, The types of scaffolds and mixing ratios corresponding to Figs. C, D, F and G. **C**, Appearance of Zein- Ca^{2+} -KGM-SA scaffolds. **D**, Appearance of Zein- Ca^{2+} -KGM-SA scaffolds after addition of medium. **E**, Focus views of the Zein- Ca^{2+} -KGM-SA scaffolds, taken in the order of ethanol replacement, medium addition to the wells, and scaffolds, are shown in the left, middle, and right images, respectively. **F**, Appearance of SPI- Ca^{2+} -KGM-SA scaffolds after addition of medium. **G**, Representative images of calcein-AM (green) and PI (red) of C2C12 cultured onto the SPI- Ca^{2+} -KGM-SA scaffold for 24 h. Scale bar, 100 μm . The SPI- Ca^{2+} -KGM scaffold collapsed structurally after 24 h of cell culture inoculation, and the three leftmost figures show its

appearance after collapse. **H**, Focus images of SPI-Ca²⁺-KGM-SA scaffolds, taken in the following order: scaffolds after ethanol replacement (left); wells after medium addition (middle); scaffolds after C2C12 inoculation and 24-hour incubation (right).

References:

1. Hasnain MS. et al. Alginates: sources, structure, and properties. In: Alginates in Drug Delivery (eds Nayak AK, Hasnain MS). *Academic Press* 1-17 (2020).
2. Shaikh MAJ, et al. Sodium alginate-based drug delivery for diabetes management: A review. *Int. J. Biol. Macromol.* **236**, 123986 (2023).
3. Pawar SN, Edgar KJ. Alginate derivatization: A review of chemistry, properties and applications. *Biomaterials* **33**, 3279-3305 (2012)
4. Elosegui-Artola A, et al. Matrix viscoelasticity controls spatiotemporal tissue organization. *Nat. Mater.* **22**, 117-127 (2023).
5. Ye S, et al. Konjac Glucomannan (KGM), Deacetylated KGM (Da-KGM), and Degraded KGM Derivatives: A Special Focus on Colloidal Nutrition. *J. Agr. Food Chem.* **69**, 12921-12932 (2021).
6. Ben-Arye, T. et al. Textured soy protein scaffolds enable the generation of three-dimensional bovine skeletal muscle tissue for cell-based meat. *Nat. Food* **1**, 210-220 (2020).
7. Pasitka L, et al. Spontaneous immortalization of chicken fibroblasts generates stable, high-yield cell lines for serum-free production of cultured meat. *Nat. Food* **4**, 35-50 (2023).
8. Xiang, N. et al. 3D porous scaffolds from wheat glutenin for cultured meat applications. *Biomaterials* **285**, 121543 (2022).
9. Zheng Y-Y. et al. Quality evaluation of cultured meat with plant protein scaffold. *Food Res. Int.* **161**, 111818 (2022).
10. Song W-J, et al. Production of cultured fat with peanut wire-drawing protein scaffold and quality evaluation based on texture and volatile compounds analysis. *Food Res. Int.* **160**, 111636 (2022).
11. Buonvino S, et al. New vegetable-waste biomaterials by Lupin albus L. as cellular scaffolds for applications in biomedicine and food. *Biomaterials* **293**, 121984 (2023).
12. Takahashi H, et al. Harvest of quality-controlled bovine myogenic cells and biomimetic bovine muscle tissue engineering for sustainable meat production. *Biomaterials* **287**, 121649 (2022).
13. Andreassen RC, et al. Production of food-grade microcarriers based on by-products from the food industry to facilitate the expansion of bovine skeletal muscle satellite cells for cultured meat production. *Biomaterials* **286**, 121602 (2022).
14. Kang, D.H. et al. Engineered whole cut meat-like tissue by the assembly of cell fibers using tendon-gel integrated bioprinting. *Nat. Commun.* **12**, 5059 (2021).

(3) Why was the confocal experiment only performed with F-Actin and DAPI? Why were other immunofluorescence staining, such as Myogenin, Desmin, or Myosin heavy chain, not included to assess the cells cultured on the 3D scaffold?

R: Thanks for your comment. Since current researches^{1,2,3} suggested that F-actin could be utilized to identify myofibers, we used F-actin and DAPI to evaluate the three-dimensional differentiation of pgEpiSCs-MCs on 3D plant scaffolds without animal-derived components. According to the suggestion, we performed additional experiments of immunofluorescence staining using Myosin and MF20, and the results showed that pgEpiSCs-MCs had good capacity to differentiate on this plant-based scaffold. This result was added to the revised manuscript (as shown in **Supplementary Fig. 12f**).

References:

1. MacQueen LA, et al. Muscle tissue engineering in fibrous gelatin: implications for meat analogs. *NPJ Sci. Food* **3**, 20 (2019).
2. Ben-Arye T, Shandalov Y, Ben-Shaul S, Landau S, Levenberg S. Textured soy protein scaffolds enable the generation of three-dimensional bovine skeletal muscle tissue for cell-based meat. *Nat. Food* **1**, 210-220 (2020).
3. Yen F-C, et al. Cultured meat platform developed through the structuring of edible microcarrier-derived microtissues with oleogel-based fat substitute. *Nat. Commun.* **14**, 2942 (2023).

(4) Based on the Calcein/PI staining, the cell morphology appears to be very round.

How can the authors ensure that they were fully proliferated and differentiated on the scaffolds?

R: Thanks for the comment. We performed additional experiments by SEM to confirm that the cells attached to the scaffolds properly and could adhere well to the scaffold. We further repeated the experiment several times with similar results and hypothesized that the cell morphology may be associated with the scaffold material, its permeability to light, or the cell type. Similar results were observed when the paper's authors incubated mouse fibroblasts (L929 cells) with USO-grafted cotton gauze¹.

Notes: The figure was cited from references ¹.

Additionally, the proliferation status of the pgEpiSCs-MCs on this plant-based scaffold was also confirmed by counting the number of cells with the nuclear localization of GFP signals on various days (as shown in **Supplementary Fig. 12a**). The number of cells significantly increased from day 1 to day 10, and the results of immunofluorescence staining (F-actin, Myosin, and MF20) revealed that pgEpiSCs-MCs underwent differentiation and multinucleated myotube formation was observed (as shown in **Fig. 5d** and **Supplementary Fig. 12f**).

References:

1. He, H., Zhou, W., Gao, J. et al. Efficient, biosafe and tissue adhesive hemostatic cotton gauze with controlled balance of hydrophilicity and hydrophobicity. *Nat. Commun.* **13**, 552 (2022).

(5) Could the authors provide zoomed-in SEM images to observe how the cells interact with the scaffolds?

R: Thanks a lot for your suggestion. We repeated the cell inoculation experiment to observe how the cells interact with the scaffolds, and this result was added to the revised manuscript (as shown in **Supplementary Fig. 12c**). The results of zoomed-in SEM images showed that pgEpiSCs-MCs were able to stably attach to the plant-based scaffolds without any animal-derived components. Cell adhesion and stretching could also be clearly seen after cell attachment to reveal a full cell shape. However, the shapes of the plant scaffolds with unattached cells showed smooth surfaces.

Notes: The interaction of pgEpiSCs-myoblast inoculated for 24 h with the scaffold was observed using SEM. Scale bar, 20 μm . Cells are indicated with red arrows, the scaffold outline without cells attached is marked by blue arrows.

(6) Apart from the amino acid profile provided in the supplementary materials, will the authors be able to provide the total protein content and other nutritional values for the pgEpiSCs-CM sample?

R: Thanks for your suggestion. We first measured the amino acid composition of pgEpiSCs-CM to evaluate its nutritional value. Additional experiments were performed on the total protein content and other nutritional values (such as fat content and trace elements) for the pgEpiSCs-CM sample with the aid of your helpful suggestions. The total protein content in pgEpiSCs-CM was around 3.92% compared to the plant-based scaffolds without cell inoculation, and trace elements (such as Zn, Ca, Mg, and K) were found. This result was added to the revised manuscript (as shown in **Supplementary Table 2** and **Lines 288-291**).

Supplementary Table 2 Analyzing the nutritional composition of pgEpiSCs-CM.

Items	Scaffold (without cells)	pgEpiSCs-CM
H ₂ O	95.82 %	95.74 %
Total Protein	No	3.92 %
Total Fat	No	No
Zinc (Zn)	156.89 $\mu\text{g/Kg}$	999.67 $\mu\text{g/Kg}$
Calcium (Ca)	14539.67 $\mu\text{g/Kg}$	1708.78 $\mu\text{g/Kg}$
Iron (Fe)	16.17 $\mu\text{g/Kg}$	5.02 $\mu\text{g/Kg}$
Magnesium (Mg)	9.76 $\mu\text{g/Kg}$	57.564 $\mu\text{g/Kg}$
Potassium (K)	0.47 $\mu\text{g/Kg}$	271.26 $\mu\text{g/Kg}$

Reviewer #2 (Remarks to the Author):

Overall this is an impressive piece of work, which spans a range of relevant topic areas in cultured meat, from cell line generation through to tissue formation. Whilst the results presented are certainly interesting, at this point in time, there are some major weaknesses in the study. The primary weakness that needs to be addressed is the demonstration of myogenic differentiation at the protein level, in both 2d and edible 3d constructs.

Please find attached some major and minor comments.

Major comments:

(1) The major weakness in this study is the lack of convincing evidence of myogenic differentiation, and a strong overreliance on bulk RNA-based data to try to make this point.

R: Thanks to the reviewer for this comment. First, we succeeded in trying to direct the differentiation of pgEpiSCs into muscle cells (MCs) without the use of serum, and the results are shown in **Fig. 2** and **its Supplementary figures**. Here, we performed transcriptome sequencing to reflect general biological features of the differentiated cells in order to clarify the feasibility of the established serum-free myogenic differentiation system. The histological data also showed that pgEpiSCs as pluripotent stem cells were differentiating into MCs, and more myogenic differentiation genes were detected. Meanwhile, we performed RT-PCR to verify the detected genes related to myogenic differentiation and collagen formation in order to confirm the validity of the transcriptome data (as shown in **Fig. 3f, g**).

In addition, we performed additional experiments to determine the expression of OCT4 (related to pluripotency) and MYH3 (related to myofiber maturation) during the myogenic differentiation of pgEpiSCs by western blot at the protein levels (as shown in **Fig. 2h, j**). Furthermore, we performed immunofluorescence staining of MF20, and our results suggested that pgEpiSCs-MCs could indeed form multinucleated myotubes with higher magnifications and quantified the fusion index (as shown in **Fig. 2f, g, i**). As well, we performed experiments to address your concern about determining whether pgEpiSCs-

MCs had endothelial or mesenchymal cell characteristics. Besides, we also analyzed the trends in gene expression associated with myogenic differentiation over time courses and discussed the long period of the differentiation process with the limitations of the CM production process in the revised manuscript. All of these results were added to the revised manuscript. Meanwhile, we attempted experiments with a variety of myofiber-related antibodies (such as MF20, Myosin, TITIN, MyHc, or Dsemin) to identify myofibrillar features in pgEpiSCs-MCs, but commercial antibodies against livestock animal species are scarce and interspecies-specific, resulting in limitations to immunofluorescence or western blot¹; therefore, we further illustrated the features of pgEpiSCs-MCs by RT-PCR and histological sequencing technology.

References:

1. Guan, X., Zhou, J., Du, G. & Chen, J. Bioprocessing technology of muscle stem cells: implications for cultured meat. *Trends Biotechnol* **40**, 721-734 (2022).

(2) In general, the differentiation images provided are of poor quality, and are not quantified in any fashion (for example using fusion index, which is very standard in the field of myogenesis). From the brightfield images, it seems like a lot of cells are aligned, but probably not fused, although it is hard to tell. The fluorescent stainings likewise show little convincing evidence of fusion, even though there is plenty of signal. No undifferentiated controls are provided for comparison, nor is there any time course data showing the process of myogenesis. These data must be substantially improved with higher quality images at higher magnifications.

R: Thanks for your comment. We repeated the differentiation experiments to improve the quality of the images and confirmed the multinucleated myotubes fused by immunofluorescence staining, as well as quantified the fusion index by MF20 immunostaining and provided undifferentiated cells as a negative control. The pgEpiSCs-derived MCs showed that multiple spindle-shaped nuclei were arranged, suggesting multinucleated myotube formation, which was further confirmed by expressing markers of mature skeletal muscle fibers (SMFs), such as Myosin, MF20, and MYH3 (as shown in **Fig. 2f-j** and **Supplementary Fig. 5a**). These results were added to the revised manuscript.

According to the suggestion about time course data showing the process of myogenesis, we collected cell samples every two days for RT-PCR to analyze the expression of relevant genes (as shown below), including key marker genes related to pluripotency (*OCT4* and *NANOG*), paraxial mesoderm differentiation (*T* and *MSGN1*), muscle stem cells (*PAX7* and *MYOD*), myogenic maturation (*MYOG*, *MYH3*, and *MYH11*), mesenchymal differentiation (*CD90* and *CD105*), and endothelial differentiation (*CD31* and *CDH5*). The results indicated that the pluripotency genes (*OCT4* and *NANOG*) were down-regulated as differentiation advanced and did not significantly change during the later differentiation process, whereas paraxial mesoderm differentiation-related genes (*T* and *MSGN1*) and muscle stem cell-related genes (*PAX7* and *MYOD*) showed a tendency for increasing and then decreasing, which also indicated directional myogenic differentiation. Meanwhile, the expression of *MYOG*, *MYH3*, and *MYH11* fluctuated a little bit at a later stage, but it remained high compared to undifferentiated cells or pgEpiSCs-derived MPCs. Along with this, we analyzed the genes associated with mesenchymal differentiation (*CD90* and *CD105*) and endothelial differentiation (*CD31* and *CDH5*). We found that the endothelial differentiation-related genes (*CD31* and *CDH5*) were not yet expressed in the myogenic differentiation system of pgEpiSCs, whereas the mesenchymal differentiation-related genes (*CD90* and *CD105*) were elevated and then decreased. This result was added to the revised manuscript (as shown in **Supplementary Fig. 6** and **Lines 187-189**).

(3) For the 3d tissues, there is again an absence of convincing differentiation data. The supplementary material (Fig. S5) does contain some myosin stainings that look promising in terms of myotube formation, although again these are provided without quantification or adequate controls for comparison. In fact, the tubes that were formed are surprisingly reminiscent of tube formation that is seen in endothelial cells and some mesenchymal cells. S5A likewise seems to show several multinucleated myotubes, but it is hard to assess by eye what proportion of cells are actually fused. Given that this point is fairly central to the author's claim to be

producing differentiated meat-like tissue, these data need to be extended, improved and emphasized more heavily in the presentation of the article.

R: Thanks for your comment. For three-dimensional differentiation, we performed additional experiments to confirm the myogenic differentiation by the assays of western blot and immunofluorescence staining for the expressions of OCT4, MYH3, F-actin, MF20 and Myosin, and quantified the results of the protein assays. The results were added to the revised manuscript (as shown in **Fig. 5d** and **Supplementary Fig. 12f, h, i**). In addition, we have provided undifferentiated cells as a negative control (as shown in **Fig. 2f**) and quantified the fusion index (**Fig. 2i**).

Additionally, we performed extra experiments to address your concern about determining whether pgEpiSCs-MCs had endothelial or mesenchymal cell characteristics, as shown below (**Response Figure 2**). pgEpiSCs were directed to differentiate into endothelial cells (ECs) and mesenchymal cell-like cells (MSCLCs) in the absence of serum (**Response Figure 2A**), using the methods in the previous study¹⁻⁵ and the cellular morphology, gene expression, and immunofluorescence staining were analyzed. We found that pgEpiSCs, pgEpiSCs-MCs, pgEpiSCs-MSCLCs, and pgEpiSCs-ECs were significantly different in morphology (**Response Figure 2B**). The differentiated cells also did not express genes related to pluripotency (*OCT4* and *NANOG*), and they expressed their own specific genes, such as pgEpiSCs-MSCLCs expressed *CD90*, *CD105*, pgEpiSCs-ECs expressed *CD31*, *CDH5*, and pgEpiSCs-MCs expressed *MYOG*, *MYMK*, *MYH2*, and *MYH3*, respectively (**Response Figure 2C**). These are consistent with the results of the RNA-Seq data (**Response Figure 2E**). Meanwhile, the results of immunofluorescence staining showed that undifferentiated pgEpiSCs expressed pluripotency markers (*OCT4* or *NANOG*), and did not express muscle cell-associated marker (*MF20*), mesenchymal stromal cell-associated marker (*PDGFR α*), and endothelial cell-associated marker (*CD31*), whereas pgEpiSCs-MCs expressed *MF20*, not *PDGFR α* and *CD31*, while pgEpiSCs-MSCLCs and pgEpiSCs-ECs also did not express *MF20* (**Response Figure 2D**). The trends in gene expression associated with myogenic differentiation over time courses were also analyzed, showing that the endothelial

differentiation-related genes (*CD31* and *CDH5*) were not yet expressed in the myogenic differentiation system of pgEpiSCs, whereas the mesenchymal differentiation-related genes (*CD90* and *CD105*) were elevated and then decreased (as shown in **Supplementary Fig. 6**). These results indicated that pgEpiSCs-MCs should not be regarded as ECs or MSCLCs.

Response Figure 2: Characterization of pgEpiSCs, pgEpiSCs-MCs, pgEpiSCs-MSCLCs, and pgEpiSCs-ECs. **A**, Schematic of serum-free directed differentiation of pgEpiSCs into MSCLCs, ECs and MCs. **B**, Cell morphology of pgEpiSCs, pgEpiSCs-MSCLCs,

pgEpiSCs-ECs and pgEpiSCs-MCs. Scale bar, 50 μ m. **C**, Expression of genes related to differentiation of pgEpiSCs as evaluated by qPCR. Pluripotency-related genes: *OCT4*, *NANOG*; Mesenchymal differentiation-related genes: *CD90*, *CD105*; Endothelial differentiation-related genes: *CD31*, *CDH5*. Error bars indicate means \pm SD with different letters in each group that differ significantly, n = 3. **D**, Immunostaining of OCT4, NANOG, PDGFR α , CD31 and MF20 in pgEpiSCs, pgEpiSCs-MCs, pgEpiSCs-MSCLCs, and pgEpiSCs-ECs. Scale bar, 20 μ m. **E**, RNA-seq data showing the expression of genes involved in pluripotency and the differentiation of mesenchymal, endothelial and myogenic. **For B-D**, MSCLCs: mesenchymal cell-like cells; ECs: endothelial cells; MCs: muscle cells.

References:

1. Orlova VV, Van dH, Francijna E, Petrus-Reurer S, Drabsch Y, Ten Dijke P, Mummery CLJNP. Generation, expansion and functional analysis of endothelial cells and pericytes derived from human pluripotent stem cells. *Nat. Protoc.* **9**, 1514-1531 (2014).
2. Wimmer RA, Leopoldi A, Aichinger M, Kerjaschki D, Penninger JM JNP. Generation of blood vessel organoids from human pluripotent stem cells. *Nat. Protoc.* **14**, 3082-3100 (2019).
3. Tran NT, Trinh QM, Lee GM, Han YM JSC, Development. Efficient differentiation of human pluripotent stem cells into mesenchymal stem cells by modulating intracellular signaling pathways in a feeder/serum-free system. *Stem Cells Dev.* **21**, 1165-1175 (2012).
4. Chen YS, Pelekanos RA, Ellis RL, Horne R, Wolvetang EJ, Fisk NM JSC TM. Small Molecule Mesengenic Induction of Human Induced Pluripotent Stem Cells to Generate Mesenchymal Stem/Stromal Cells. *Stem Cells Transl. Med.* **1**, 83-95 (2012).
5. Chin CJ, et al. Transcriptionally and Functionally Distinct Mesenchymal Subpopulations Are Generated from Human Pluripotent Stem Cells. *Stem Cell Reports* **10**, 436-446 (2018).

(4) For both 2d and 3d, there are no robustly quantifiable protein measurements, for example by western blot for major muscle proteins, which needs to be addressed.

A favourable comparison with traditional meat would not be required for publication, but certainly a strong induction relative to undifferentiated cells would be expected.

R: Thanks for your suggestion. We agree with the reviewer's comment, and the muscle protein marker (MYH3) and the pluripotency marker (OCT4) were measured by western blot to demonstrate the induction of 2D and 3D at the protein level. The results showed that the expression of MYH3 was significantly increasing in pgEpiSCs-MCs in comparison to undifferentiated cells, which quantified the results of the protein assays in 2D or 3D differentiation. We have added these results to the revised manuscript (as shown in **Lines 181-185, 277, Fig. 2h, j and Supplementary Fig. 12h, i**).

(5) The RNAseq and metabolomics data are impressive, but I somewhat question the value of these data (particularly the extent to which they are presented, as two entire figures) when there is no real way to quantify these changes relative to a differentiated sample. The metabolomics data add very little to the central statement of the study. The “crucial pathways”, mentioned on line 237 for example requires much more context to be relevant for the story.

R: Thanks to the reviewer for this comment. As in reply to Comment 1, we performed transcriptome and untargeted metabolomic sequencing to reflect general biological features of the differentiated cells in order to clarify the feasibility of the established serum-free myogenic differentiation system. Our results indicated that the serum-free myogenic differentiation of pgEpiSCs is involved in metabolic pathways such as vitamin B6, the pentose phosphate pathway (PPP), mineral absorption, protein digestion and absorption.

1) Vitamin B6, as a key cofactor for various biochemical reactions in basal cellular metabolism, has an important role in muscle production^{1, 2}. Besides, as a dietary nutrient additive component, it promotes muscle development and regeneration in the organism², which also suggests that the vitamin B6 metabolic pathway plays an important role in the process of muscle development.

2) The PPP is a glucose metabolism process³. Myogenic differentiated cells may change their biosynthesis rate, resulting in lower amounts of chemicals linked to glycolysis and the PPP, which implies that myogenic differentiation depends on glycolytic metabolites⁴.

3) Mineral absorption affects myogenic differentiation by influencing Ca²⁺, Mg²⁺, and other ions. Mg²⁺ promotes myogenic differentiation in C2C12 and aged MuSCs⁵, Ca²⁺ also accelerates myogenic differentiation and myotube formation⁶, which suggests that mineral uptake has an important effect on myogenic differentiation.

4) Protein digestion and absorption are also able to influence myogenic differentiation. Tryptophan-fed mice could enhance myogenic differentiation and myotube maturation^{2, 7}, and the tryptophan metabolism was upregulated in late differentiation⁸. Meanwhile, arginine can accelerate myogenic differentiation and myotube formation through

synergistic Ca²⁺ action⁶.

Although a clear mechanism of action for these pathways has not yet been described, this is sufficient to suggest that myogenic differentiation is associated with these pathways. In addition, glycolysis plays an important role in myogenic differentiation^{3, 4} and we further validated the involvement of glycolysis-related genes at the RT-PCR level. In the field of skeletal muscle research and the meat industry, it has been suggested that metabolomics is not only a key technique for characterizing meat composition and investigating biomarkers for quality control but can also be used in a way that will help to optimize the composition of the culture medium during myogenic differentiation⁹. Meanwhile, these data help to understand the metabolic behavior of MCs during serum-free induction and offer a possible technical approach to further optimize the formulation of the medium by subsequently controlling the metabolic level of glucose and other metabolites, which were described in the discussion of the revised manuscript (as shown in **Lines 332-337**).

References:

1. Cellini, B.; Montioli, R.; Oppici, E.; Astegno, A.; Borri Voltattorni, C. The chaperone role of the pyridoxal 5'-phosphate and its implications for rare diseases involving B6-dependent enzymes. *Clin. Biochem.* **47**, 158–165 (2014).
2. Kumar A, Kumar Y, Sevak JK, Kumar S, Kumar N, Gopinath SD. Metabolomic analysis of primary human skeletal muscle cells during myogenic progression. *Sci. Rep.* **10**, 11824 (2020).
3. Jang, M., Scheffold, J., Røst, L.M. et al. Serum-free cultures of C2C12 cells show different muscle phenotypes which can be estimated by metabolic profiling. *Sci. Rep.* **12**, 827 (2022).
4. Wüst S, et al. Metabolic Maturation during Muscle Stem Cell Differentiation Is Achieved by miR-1/133a-Mediated Inhibition of the Dlk1-Dio3 Mega Gene Cluster. *Cell Metabolism* **27**, 1026-1039.e1026 (2018).
5. Liu Y, et al. Magnesium supplementation enhances mTOR signalling to facilitate myogenic differentiation and improve aged muscle performance. *Bone* **146**, 115886 (2021).
6. Gong L, Zhang X, Qiu K, He L, Wang Y, Yin J. Arginine promotes myogenic differentiation and myotube formation through the elevation of cytoplasmic calcium concentration. *Animal Nutrition* **7**, 1115-1123 (2021).
7. Sakuma, K., Aoi, W. & Yamaguchi, A. The intriguing regulators of muscle mass in sarcopenia and muscular dystrophy. *Front. Aging Neurosci.* **6**, 230 (2014).
8. Lemons, J. M. S. et al. Quiescent fibroblasts exhibit high metabolic activity. *PLoS Biol.* **8**, e1000514 (2010).

9. Messmer, T. et al. A serum-free media formulation for cultured meat production supports bovine satellite cell differentiation in the absence of serum starvation. *Nat. Food* **3**, 74-85 (2022).

(6) The differentiation scheme seems unusually long, given that most myogenic differentiation protocols take place over a matter of days. Some timecourse data to investigate whether this process is really required might be interesting. At a bare minimum, the authors should discuss this point and the potential limitations that it might have for an cultured meat production process based on this system.

R: Thanks to the reviewer's comment. pgEpiSCs are a type of pluripotent stem cell that has the capacity to differentiate into any type of cell. During the differentiation process, the pgEpiSCs go through the stages of paraxial mesoderm, muscle progenitor cells (MPCs), myoblast, and myoblast fusion. It should take longer for differentiation into MCs than MuSCs or C2C12, and a similar phenomenon has been observed in pluripotent stem cells of both humans¹ and mice².

We collected cell samples every two days for RT-PCR to analyze the expression of relevant genes about time course data showing the process of myogenesis (as shown in **Supplementary Fig. 6** and mentioned in Comment 2), including key marker genes related to pluripotency (*OCT4* and *NANOG*), paraxial mesoderm differentiation (*T* and *MSGN1*), muscle stem cells (*PAX7* and *MYOD*), myogenic maturation (*MYOG*, *MYH3*, and *MYH11*), mesenchymal differentiation (*CD90* and *CD105*), and endothelial differentiation (*CD31* and *CDH5*). The results indicated that the pluripotency genes (*OCT4* and *NANOG*) were down-regulated as differentiation advanced and did not significantly change during the later differentiation process, whereas paraxial mesoderm differentiation-related genes (*T* and *MSGN1*) and muscle stem cell-related genes (*PAX7* and *MYOD*) showed a tendency for increasing and then decreasing, which also indicated directional myogenic differentiation. Meanwhile, the expression of *MYOG*, *MYH3*, and *MYH11* fluctuated a little bit at a later stage, but it remained high compared to undifferentiated cells or pgEpiSCs-derived MPCs. Along with this, we analyzed the genes associated with mesenchymal differentiation (*CD90* and *CD105*) and endothelial differentiation (*CD31* and *CDH5*). We found that the endothelial differentiation-related genes (*CD31* and *CDH5*) were not yet expressed in the

myogenic differentiation system of pgEpiSCs, whereas the mesenchymal differentiation-related genes (*CD90* and *CD105*) were elevated and then decreased.

Meanwhile, the fact that pgEpiSCs keep proliferating during a longer period of differentiation indicates that they can generate a larger cell number, but the myogenic differentiation process of pgEpiSCs is more complex than that of MuSCs or immortalized myoblast cell lines (such as bovine satellite cells³, chicken fibroblasts or C2C12⁴), which needs to be simplified or standardized to meet the requirements for the production process of CM in the future and was described in the discussion of the revised manuscript (as shown in **Lines 361-366**).

References:

1. Wu, J. *et al.* A Myogenic Double-Reporter Human Pluripotent Stem Cell Line Allows Prospective Isolation of Skeletal Muscle Progenitors. *Cell Rep.* **25**, 1966-1981 e1964 (2018).
2. Darabi, R. *et al.* Human ES- and iPS-derived myogenic progenitors restore DYSTROPHIN and improve contractility upon transplantation in dystrophic mice. *Cell Stem Cell* **10**, 610-619 (2012).
3. Stout AJ, *et al.* Immortalized Bovine Satellite Cells for Cultured Meat Applications. *ACS Synth. Biol.* **12**, 1567-1573 (2023).
4. Pasitka, L. *et al.* Spontaneous immortalization of chicken fibroblasts generates stable, high-yield cell lines for serum-free production of cultured meat. *Nat. Food*, **4**, 35-50 (2023).

(7) It is difficult to assess differentiation efficiency (percentage of cells/clones that participate in differentiation) as only level of differentiation is shown and compared to literature data, not positive controls (e.g. C2C12s).

R: Thank you for your comment. Here, we determined the efficiency of pgEpiSCs to differentiate into MPCs. We also performed experiments to evaluate the differentiation effectiveness of pgEpiSCs-MPCs in accordance with your suggestion, utilizing C2C12 as a positive control, and the results were shown below (**Response Figure 3**). Firstly, the results of immunofluorescence suggested that C2C12 expressed MYOD and PAX7 was negatively expressed (**Response Figure 3A**), in contrast to pgEpiSCs-MPCs, which expressed PAX7 and MYOD simultaneously. Next, the proportion of CD31⁺CD45⁺CD56⁺ was only 0.14% for C2C12, according to the results of the flow cytometry, whereas the

percentage of pgEpiSCs-MPCs was positive up to 99.8% (**Response Figure 3B**). Furthermore, co-expression of PAX7 and MYOD is a characteristic of muscle stem cells (MuSCs), while CD56 is also a marker of MuSCs¹. These results indicated that pgEpiSCs-MPCs possessed the characteristics of MuSCs, whereas C2C12, as an immortalized myoblast cell line, does not have the characteristics of MuSCs. In addition, we chose to compare the myogenic differentiation efficiency of pluripotent stem cells with that of humans and mice since there are no pluripotent stem cell lines that can be passed on stably for a long time *in vitro* in livestock species (such as pigs).

Response Figure 3: Comparative analysis of differentiation efficiency of pgEpiSCs-MPCs. **A**, Immunostaining of PAX7 and MYOD in C2C12 and pgEpiSCs-MPCs. Scale bar, 20 μ m. **B**, Flow cytometric analysis of the proportion of CD31-CD45-CD56⁺ cell populations in C2C12 and pgEpiSCs-MPCs.

References:

1. Brunet, A., Goodell, M.A. & Rando, T.A. Ageing and rejuvenation of tissue stem cells and their niches. *Nat. Rev. Mol. Cell Biol.* **24**, 45–62 (2023).

(8) Scaffold: Please show cell adhesion to scaffold material.

R: Thanks for your sincere comment. We re-inoculated cells with plant-based scaffolds composed of KGM and SA without the addition of any animal-derived components or chemical alterations to enhance cell adhesion. Subsequently, we utilized SEM for

observation in order to better show cell adhesion to scaffold material, which was added to the revised manuscript (as shown in **Supplementary Fig. 12c**). The results demonstrated that cells could show their shape and adhere well to the plant-based scaffolds. The plant scaffolds without connected cells, however, had a smooth surface in their form.

Notes: The interaction of pgEpiSCs-myoblast inoculated for 24 h with the scaffold was observed using SEM. Scale bar, 20 μm . Cells are indicated with red arrows, the scaffold outline without cells attached is marked by blue arrows.

(9) Line 175: the corresponding figure says 99.8%.

R: Thanks to the reviewer for drawing our attention to this mistake, and we apologize for the misdescription of the results. The proportion of pgEpiSCs that had differentiated into MPCs was evaluated by using flow cytometry, and the percentage of CD56⁺ cells from the CD31⁻CD45⁻ cell population was analyzed. The results suggested that the percentage of CD31⁻CD45⁻ cells in pgEpiSCs-MPCs was 95.8%, and the percentage of CD56⁺ cells in this population was as high as 99.8%. Following the reviewer's suggestion, we have rewritten it and highlighted it in red in **Line 178**.

(10) Supplementary Fig 9H y-axis labels are missing.

R: Thanks to the reviewer for pointing out the problem. We apologize for the mistake we made in processing the images of the manuscript, since the graph's center y-axis was missing. Furthermore, the y-axis was added in **Supplementary Fig. 10h**.

(11) Supplementary Fig 10D, please provide stress Pa, to provide reference to muscle tissue.

R: Thanks for this comment from the reviewer. Following the reviewer's suggestion, we have changed the unit of stress from g to Pa with a red highlight in **Supplementary Fig. 11d**, which was in keeping with your suggestion for providing reference to muscle tissue.

(12) Line 316. There are several studies in literature proving this wrong.

R: Thanks to your comment for pointing out this mistake. The sentence had been revised and highlighted in red in the revised manuscript (as shown in **Lines 322-326**).

Minor comments

(13) "Currently accessible cell lines include mesenchymal stem cells and muscle stem cells (MuSCs), which can be amplified only a limited number of times before losing their capacity to differentiate."

This point in the introduction is not really the case, as immortalised adult stem cells, such as bovine satellite cells, are now available (Stout et al). Such lines are also available for avian species, such as chicken.

R: Thanks for pointing out our shortcomings. After reviewing the relevant studies, we have revised the original sentence and highlighted it in red (as shown in **Lines 79-84**).

(14) It is rather hard to know what to make of the karyotype analysis from the provided figure(s), given that no quantification nor labelling is provided.

R: Thanks a lot for your comment. Pigs typically have 38 chromosomes. Our results showed that multiple generations of pgEpiSCs, pgEpiSCs-MCs, and subsequent differentiation of this cell in 3D scaffolds without animal-derived components retained the usual chromosomal number of 38, which indicated that long-term culture, serum-free myogenic differentiation, and three-dimensional differentiation of pgEpiSCs did not result in the occurrence of chromosome number variation. Based on this, we rearranged the karyotype figures for easier comprehension in the revised manuscript (as shown in **Supplementary Fig.1g, Supplementary Fig. 5d and Supplementary Fig.12e**).

(15) The article and figures need to be thoroughly checked over for English spelling and grammar. F-actin should be spelt with a hyphen. 'Cancer hallmarkers' does not make sense as a graph title, and there are many other examples.

R: Thanks for your comment. We sincerely apologize for the English spelling and grammar errors in the manuscript and have addressed each of these problems in the revised manuscript. Firstly, the "F-actin" has been written with a hyphen (as shown in **Fig. 5d**). Additionally, we realized that it was not suitable to use "cancer markers" as the graph title, so we changed it and emphasized the meaning of the graph (as shown in **Supplementary Fig. 3f**). Furthermore, we updated the graphic title in **Fig. 2e** to "The efficiency of differentiation (%)". In addition, we have checked the English spelling and grammar with a red highlight in the revised manuscript.

(16) Some of the graph axes are rather strangely named, for example 'The damage levels of DNA'.

R: Thanks for your comment. The alkaline comet assay is a more sensitive and effective method for assessing DNA damage. We intend on using it to measure the amounts of DNA damage in multiple generations of pgEpiSCs, and the cells that have been exposed to H₂O₂ as a positive control group. We also realized that it is not suitable to use "The damage levels of DNA" as the graph axes after you voiced this concern, and we changed the name of the graph axes to "The quantification of comets" and presented them with red highlighted in the revised manuscript (as shown in **Supplementary Fig. 1i**).

Reviewer #3 (Remarks to the Author):

I co-reviewed this manuscript with one of the reviewers who provided the listed reports as part of the Nature Communications initiative to facilitate training in peer review and appropriate recognition for co-reviewers.

R: Thank you very much for co-reviewing this manuscript.

Other changes:

- Following the reviewers' suggestions, we updated the pertinent experimental results and revised the figures, as well as further refined the methods section based on the newly added experiments' results, which are highlighted in red in the revised manuscript.
- Following the reviewers' comments, we further revised the discussion section.
- As a result of responding reviewers' comments, the references have been rearranged with red highlights in the revised manuscript.
- We added a note on the availability of metabolome data and highlighted it in red in the revised manuscript (as shown in **Line 955-956**).
- New primer sequences were added due to the addition of new experimental data and are highlighted in red (as shown in **Supplementary Table 1**).
- The serial numbers in the figure were changed from uppercase to lowercase in accordance with the formatting instructions for *Nature Communications*.

We would like to thank the reviewers once again for their time and effort in providing helpful suggestions on our manuscript.

REVIEWERS' COMMENTS

Reviewer #1 (Remarks to the Author):

The authors clarified most of my comments except for the cell morphology parts. The SEM images do not clearly illustrate that the cells can fully spread on the scaffolds. The cells appear to be collapsing on the scaffold, which might be a result of the fixation and dehydration process. Thus, I recommend that the authors perform Actin or MHC staining and zoom in to X20 or X40 magnification to observe cell morphology.

Reviewer #2 (Remarks to the Author):

The revision shows a large number of additional experiments and data on top of the already extensive dataset presented in the original. The description of the different cell lineages derived from the porcine epiblasts is now quite convincing. Also, the 2D differentiation into muscle is up to standard. 3D differentiation is still a bit hard to qualify, but overall it seems adequate and meets the standard that is average for the field.

One concerning remark in the rebuttal (response to q7 of reviewer2), which I am not sure was there in the original manuscript is that there are not pluripotent stem cell lines that can be stably passed on for livestock species. It is not clear if this is a general statement or it also includes pgEpiSCs? In the original text, the authors state that they maintain cultures in undifferentiated state for 200 passages. Should I conclude that the previous statement does NOT include pgEpiSCs?

Reviewer #3 (Remarks to the Author):

RE: “Generation of three-dimensional meat-like tissue from stable pig epiblast stem cells” (NCOMMS-23-05852C)

Point-by-point response to reviewers’ comments:

REVIEWER COMMENTS

Reviewer #1 (Remarks to the Author):

The authors clarified most of my comments except for the cell morphology parts. The SEM images do not clearly illustrate that the cells can fully spread on the scaffolds. The cells appear to be collapsing on the scaffold, which might be a result of the fixation and dehydration process. Thus, I recommend that the authors perform Actin or MHC staining and zoom in to X20 or X40 magnification to observe cell morphology.

R: Thanks to the reviewer for this suggestion. Following your suggestion, we performed Myosin staining and utilized a confocal microscope to observe cell morphology (X40, as shown below). The results of zooming in to X40 magnification showed that cells could stably attach to the plant-based scaffolds to reveal a full cell shape.

Response Figure. Confocal observation of the affinity state of cells with the scaffold and immunofluorescence staining of Myosin (red), scale bar, 20 μ m.

Reviewer #2 (Remarks to the Author):

The revision shows a large number of additional experiments and data on top of the already extensive dataset presented in the original. The description of the different cell lineages derived from the porcine epiblasts is now quite convincing. Also, the 2D differentiation into

muscle is up to standard. 3D differentiation is still a bit hard to qualify, but overall it seems adequate and meets the standard that is average for the field.

One concerning remark in the rebuttal (response to q7 of reviewer2), which I am not sure was there in the original manuscript is that there are not pluripotent stem cell lines that can be stably passed on for livestock species. It is not clear if this is a general statement or it also includes pgEpiSCs? In the original text, the authors state that they maintain cultures in undifferentiated state for 200 passages. Should I conclude that the previous statement does NOT include pgEpiSCs?

R: Thanks for your comment. Searching through the pertinent literature, it is found that pluripotent stem cells from livestock species (such as pigs¹⁻⁷, cows^{4, 8-10}, sheep^{4, 11-19}, etc.) had previously been reported, and as you mentioned, no stable pluripotent stem cell line with passage beyond 200 generations and clarification of pluripotency features had been reported before the publication of the research findings on pgEpiSCs²⁰. Based on the pgEpiSCs that we had previously established in our lab²⁰, the cell line's stability, capacity for differentiation, a serum-free myogenic differentiation system, and three-dimensional scaffolds without animal components for the development of cultured meat were further explored in this study.

References:

1. Baek S, et al. Effects of Culture Dimensions on Maintenance of Porcine Inner Cell Mass-Derived Cell Self-Renewal. *Mol. cells* **40**, 117-122 (2017).
2. Gao X, et al. Establishment of porcine and human expanded potential stem cells. *Nat. Cell Biol.* **21**, 687-699 (2019).
3. Hou DR, et al. Derivation of Porcine Embryonic Stem-Like Cells from In Vitro-Produced Blastocyst-Stage Embryos. *Sci. Rep.* **6**, 25838 (2016).
4. Kinoshita M, et al. Pluripotent stem cells related to embryonic disc exhibit common self-renewal requirements in diverse livestock species. *Development* **148**, dev199901 (2021).
5. Vassiliev I, Nottle MB. Isolation and culture of porcine embryonic stem cells. *Meth. Mol. Bio.* (Clifton, NJ) **1074**, 85-95 (2013).
6. Xue B, et al. Porcine Pluripotent Stem Cells Derived from IVF Embryos Contribute to Chimeric Development In Vivo. *PLoS One* **11**, e0151737 (2016).
7. Yuan Y, et al. A six-inhibitor culture medium for improving naïve-type pluripotency of porcine pluripotent stem cells. *Cell Death Discov.* **5**, 104 (2019).

8. Zhao L, et al. Establishment of bovine expanded potential stem cells. *Proc. Natl. Acad. Sci. U. S. A.* **118**, e2018505118 (2021).
9. Wang Y, et al. Establishment of bovine trophoblast stem cells. *Cell Rep.* **42**, 112439 (2023).
10. Su Y, et al. Establishment of Bovine-Induced Pluripotent Stem Cells. *Int. J. Mol. Sci.* **22**, 10489 (2021).
11. Li Y, Cang M, Lee AS, Zhang K, Liu D. Reprogramming of sheep fibroblasts into pluripotency under a drug-inducible expression of mouse-derived defined factors. *PLoS One* **6**, e15947 (2011).
12. Bao L, et al. Reprogramming of ovine adult fibroblasts to pluripotency via drug-inducible expression of defined factors. *Cell Res.* **21**, 600-608 (2011).
13. Song H, et al. Induced pluripotent stem cells from goat fibroblasts. *Mol. Rep. Dev.* **80**, 1009-1017 (2013).
14. Sandmaier SE, et al. Generation of induced pluripotent stem cells from domestic goats. *Mol. Rep. Dev.* **82**, 709-721 (2015).
15. Ren J, et al. Generation of hircine-induced pluripotent stem cells by somatic cell reprogramming. *Cell Res.* **21**, 849-853 (2011).
16. Liu J, Balehosur D, Murray B, Kelly JM, Sumer H, Verma PJ. Generation and characterization of reprogrammed sheep induced pluripotent stem cells. *Theriogenology* **77**, 338-346.e331 (2012).
17. Behboodi E, et al. Establishment of goat embryonic stem cells from in vivo produced blastocyst-stage embryos. *Mol. Rep. Dev.* **78**, 202-211 (2011).
18. Vilarino M, et al. Derivation of sheep embryonic stem cells under optimized conditions. *Reproduction* **160**, 761-772 (2020).
19. Liu F, et al. Derivation of Arbas Cashmere Goat Induced Pluripotent Stem Cells in LCDM with Trophectoderm Lineage Differentiation and Interspecies Chimeric Abilities. *Int. J. Mol. Sci.* **24**, 14728 (2023).
20. Zhi M, et al. Generation and characterization of stable pig pregastrulation epiblast stem cell lines. *Cell Res* **32**, 383-400 (2022).

Reviewer #3 (Remarks to the Author):

R: Thank you very much for co-reviewing this manuscript.

With that, we would like to thank you once again for your time and effort on our manuscript.